# Integrative study of skeletal muscle mitochondrial dysfunction in a murine pancreatic cancer-induced cachexia model

Tristan Gicquel[1,2], Fabio Marchiano[3], Gabriela Reyes-Castellanos[1,2], Stephane Audebert[1], Luc Camoin[1], Bianca H Habermann[3], Benoit Giannesini[4], Alice Carrier[1,2]*

[1]Aix Marseille Univ, CNRS, INSERM, Institut Paoli-Calmettes, CRCM, Marseille, France; [2]Nutrition And Cancer Research Network (NACRe Network), Jouy-en-Josas, France; [3]Aix Marseille Univ, CNRS, IBDM, Marseille, France; [4]Aix Marseille Univ, CNRS, CRMBM, Marseille, France

*For correspondence:
alice.carrier@inserm.fr

Competing interest: The authors declare that no competing interests exist.

**Abstract** Pancreatic ductal adenocarcinoma (PDAC), the most common pancreatic cancer, is a deadly cancer, often diagnosed late and resistant to current therapies. PDAC patients are frequently affected by cachexia characterized by muscle mass and strength loss (sarcopenia) contributing to patient frailty and poor therapeutic response. This study assesses the mechanisms underlying mitochondrial remodeling in the cachectic skeletal muscle, through an integrative exploration combining functional, morphological, and omics-based evaluation of gastrocnemius muscle from KIC genetically engineered mice developing autochthonous pancreatic tumor and cachexia. Cachectic PDAC KIC mice exhibit severe sarcopenia with loss of muscle mass and strength associated with reduced muscle fiber's size and induction of protein degradation processes. Mitochondria in PDAC atrophied muscles show reduced respiratory capacities and structural alterations, associated with deregulation of oxidative phosphorylation and mitochondrial dynamics pathways. Beyond the metabolic pathways known to be altered in sarcopenic muscle (carbohydrates, proteins, and redox), lipid and nucleic acid metabolisms are also affected. Although the number of mitochondria per cell is not altered, mitochondrial mass shows a twofold decrease and the mitochondrial DNA threefold, suggesting a defect in mitochondrial genome homeostasis. In conclusion, this work provides a framework to guide toward the most relevant targets in the clinic to limit PDAC-induced cachexia.

## eLife assessment

This **useful** study uses a mouse model of pancreatic cancer to examine mitochondrial mass and structure in atrophying muscle along with aspects of mitochondrial metabolism in the same tissue. Most relevant are the **solid** transcriptomics and proteomics approaches to map out related changes in gene expression networks in muscle during cancer cachexia.

## Introduction

Pancreatic cancer has a very poor prognosis and the lowest 5-year relative survival rate (11%) (**Siegel et al., 2022**). It is the third-leading cause of cancer deaths and is expected to become the second by 2026 due to its increasing incidence and resistance to therapies (**Nevala-Plagemann et al., 2020**; **Rahib et al., 2021**). Pancreatic ductal adenocarcinoma (PDAC) is the most common form of pancreatic cancer. More than 80% of PDAC patients are cachectic at the time of diagnosis, having lost body weight due to melting of metabolic organs, adipose tissue, and muscles (**Hendifar et al., 2018**).

Cancer-induced cachexia is a frequent debilitating muscle wasting syndrome characterized by lean body mass loss, muscle and adipose tissues loss, altered body metabolism, and inflammation (*Baracos et al., 2018*). It is often irreversible and a sure sign of increased mortality and morbidity (*Fearon et al., 2011*; *Stubbins et al., 2020*). PDAC-induced cachexia contributes significantly to the overall weakness of patients which hinders their clinical management and increases morbidity and mortality (*Mintziras et al., 2018*). In this context, the limitation of patients' cachexia through nutritional support - diet and physical activity - is an approach increasingly developed in cancer hospitals. However, the development of nutritional strategies is limited by the poor knowledge of the mechanisms linking nutrition, cachexia, and cancer. In addition, the knowledge of biomarkers to identify the patients that would benefit from nutritional strategies is required.

Exploration of the mechanisms underlying muscle wasting (sarcopenia) is done mostly in the aging context, nevertheless some insights were obtained in the recent years in cancer cachexia (*Argilés et al., 2014*; *Argilés et al., 2015*; *Argilés et al., 2019*; *Baracos et al., 2018*; *Biswas and Acharyya, 2020*). As in aging, the main mechanism underlying cancer sarcopenia is alterations in amino acid and protein metabolism, comprising increased muscle protein degradation (proteolysis) and impaired muscle protein synthesis. The tumor-associated systemic inflammation is one of the cancer characteristics promoting the activation of protein catabolism in skeletal muscle (*VanderVeen et al., 2017*). Dysfunction of mitochondria also contributes to changes in muscle metabolism in cancer-induced cachexia in patients and mouse models (*Beltrà et al., 2021*; *de Castro et al., 2019*; *Dolly et al., 2022*; *van der Ende et al., 2018*; *VanderVeen et al., 2017*). As the energy powerhouses of cells, mitochondria are instrumental in body organs homeostasis and function in general, and in skeletal muscle in particular since contraction to assure mobility is highly energy consuming. Beyond ATP production, mitochondria are a hub in nutrients metabolism, cell signaling, and cell death, thus controlling cell life-death balance. Dysfunctional mitochondria are observed in many pathologies that are not restricted to muscular diseases (*Annesley and Fisher, 2019*; *Murphy and Hartley, 2018*; *Russell et al., 2020*). In cancer cachexia, alterations in quantity and quality of mitochondria were reported in sarcopenic muscle, due to decreased biogenesis, increased mitophagy, altered dynamics (fusion and fission), and reduced energy metabolism through impairment of mitochondrial metabolism and respiration (*Beltrà et al., 2021*). This latter promotes production of reactive oxygen species (ROS) by the respiratory chain, increasing the inflammation-associated oxidative stress which damages mitochondria, leading to a vicious circle (*Penna et al., 2020*; *Zhang et al., 2023*).

Despite the central role of mitochondria in the physiopathology of skeletal muscle, their involvement in PDAC-induced muscle wasting was not addressed to date. Furthermore, their role in cancers gained increased attention only recently, with the now-well admitted demonstration of their key role in cancer cell metabolic reprogramming, aberrant proliferation, survival, and resistance to therapies. Our laboratory showed that mitochondrial metabolism is reprogrammed in PDAC tumor cells and constitutes a vulnerability opening novel therapeutic avenues (*Abdel Hadi et al., 2021*; *Masoud et al., 2020*; *Reyes-Castellanos et al., 2023*; *Reyes-Castellanos et al., 2020*). Nevertheless, mitochondria still have to be explored in the tumor microenvironment and distant metabolic organs. Skeletal muscle inflammation is a common feature observed in PDAC mouse models with systemic low-grade inflammation (*Gilabert et al., 2014*; *Michaelis et al., 2017*; *Zhu et al., 2019*). We thus made the hypothesis that the chronic inflammatory state induced by the pancreatic tumor could impact on skeletal muscle potentially through mitochondrial stress and dysfunction.

The objective of the present preclinical study was to investigate the mitochondrial remodeling in PDAC muscle, and decipher the underlying molecular mechanisms. Our study focused on the gastrocnemius muscle of the KIC genetically engineered mice developing spontaneously a PDAC associated with cachexia (*Aguirre et al., 2003*), by implementing different technical approaches to provide a comprehensive description of PDAC-associated mitochondrial features in sarcopenic muscle. This study led to the demonstration of profound alterations of mitochondrial function, structure, and gene expression in PDAC muscle, that could be used in the future to identify new PDAC-associated cachexia biomarkers and targets in the clinic.

# Results

## The sarcopenia is severe in cachectic KIC PDAC mice

KIC mice, in which the constitutively active mutant *Kras* oncogene is expressed and the tumor suppressor *Ink4a* is deleted, both specifically in the pancreas, spontaneously develop PDAC and associated cachexia within 9–11 weeks of age (*Aguirre et al., 2003*). Our study focused on the gastrocnemius skeletal muscle of male mice, via several technical approaches shown in *Figure 1A* (noninvasive multimodal magnetic resonance [MR] in vivo, and oxygraphy, microscopy [visible, fluorescence, and transmission electron], RNA sequencing, and protein mass spectrometry ex vivo).

KIC and littermate control mice were sacrificed and analyzed at the age when the KIC showed signs of disease (pancreatic tumor), i.e., weight loss (*Figure 1B*), kyphosis, and loss of mobility. The presence of a pancreatic tumor was confirmed by macroscopic and histological examination of the pancreas (*Figure 1C*). The weight of the different organs of interest was compared in between the control and KIC mice groups (*Figure 1D*), showing a weight gain for the KIC pancreas in which the normal pancreas was almost completely replaced by the tumor, and splenomegaly reflecting inflammation. Conversely, gastrocnemius muscle, liver, and gonadal adipose tissue showed decreased weight in KIC mice with an almost complete disappearance of adipose tissue, related to tumor-induced cachexia.

Histological analysis of gastrocnemius muscle transversal sections showed decreased overall cross-sectional area in KIC mice compared to controls (*Figure 2A*). Immunohistofluorescence microscopy demonstrated muscle cells (fibers) shrinkage with decreased fiber diameter (reflecting cross-sectional area) without modification of the total number of fibers per muscle section (*Figure 2B* and *Figure 2—figure supplement 1A*), and with a moderate change of the proportion of the different fiber types (*Figure 2—figure supplement 1B*). Sarcopenia was further evidenced by the accumulation of the Muscle RING-Finger protein-1 (MuRF1) E3 ubiquitin ligase involved in proteolysis driving atrophy, by immunohistofluorescence microscopy (*Figure 2C*) and western blotting (*Figure 2D*) analyses. These data show a pronounced sarcopenia in KIC mice associated with decrease in fiber's size and increased proteolysis.

## Noninvasive MR investigation shows altered gastrocnemius muscle function

Gastrocnemius muscle function and bioenergetics were investigated longitudinally using noninvasive multimodal MR measurements. We found that muscle volume calculated from MR images was lower in KIC mice at 9–11 weeks of age (bearing a tumor) when compared to age-matched controls (21% decrease) as well as to 7- to 8-week-old (pre-tumoral) KIC mice (*Figure 3A*). This muscle volume loss, which is similar to weight loss (–28%; *Figure 1D*), was associated with an almost proportional reduction of the absolute tetanic force which represents the maximal muscle capacity for producing force (–16%; *Figure 3A*). The specific tetanic force, calculated by scaling the absolute tetanic force to muscle volume, did not differ between KIC and controls at 9–11 weeks of age (*Figure 3A*), hence indicating that the reduction in absolute tetanic force was fully accounted by muscle loss. In addition, there was no difference between phenotypes and ages for the twitch force-generating capacity during the 6 min bout of exercise and the end-exercise twitch force level (*Figure 3B*). Overall, these data demonstrate that muscle quality was not altered in KIC mice bearing tumor. In contrast, the twitch half-relaxation time averaged over the whole 6 min exercise was longer in KIC mice bearing a tumor when compared to age-matched controls and to 7- to 8-week-old (pre-tumoral) KIC mice (*Figure 3B*), thereby suggesting that muscle biomechanical properties were disturbed in KIC cancer mice.

We then used dynamic $^{31}$P-MR spectroscopy for continuously assessing in vivo muscle bioenergetics before (rest), during, and after the 6 min bout of exercise. The corresponding changes in phosphocreatine (PCr) and pH over the course of time and quantifications are shown in *Figure 3C and D*. At rest, there were no differences among phenotypes and ages for PCr level and pH, which indicates that the presence of tumor did not affect the basal bioenergetics status. However, we found that the time constant of post-exercise PCr resynthesis ($\tau$PCr) was 43% longer in KIC bearing a tumor compared to age-matched controls. Given that $\tau$PCr during the post-exercise recovery relies exclusively on oxidative ATP synthesis (*Kemp et al., 2015*), the longer $\tau$PCr reveals that the presence of tumor did alter the mitochondrial function in vivo. Accordingly, we found that the maximal oxidative capacity, calculated as the ratio between basal PCr and $\tau$PCr (*Prompers et al., 2014*), was lower in KIC mice (–31%). It should be noted that the end-exercise PCr level, which reflects the metabolic

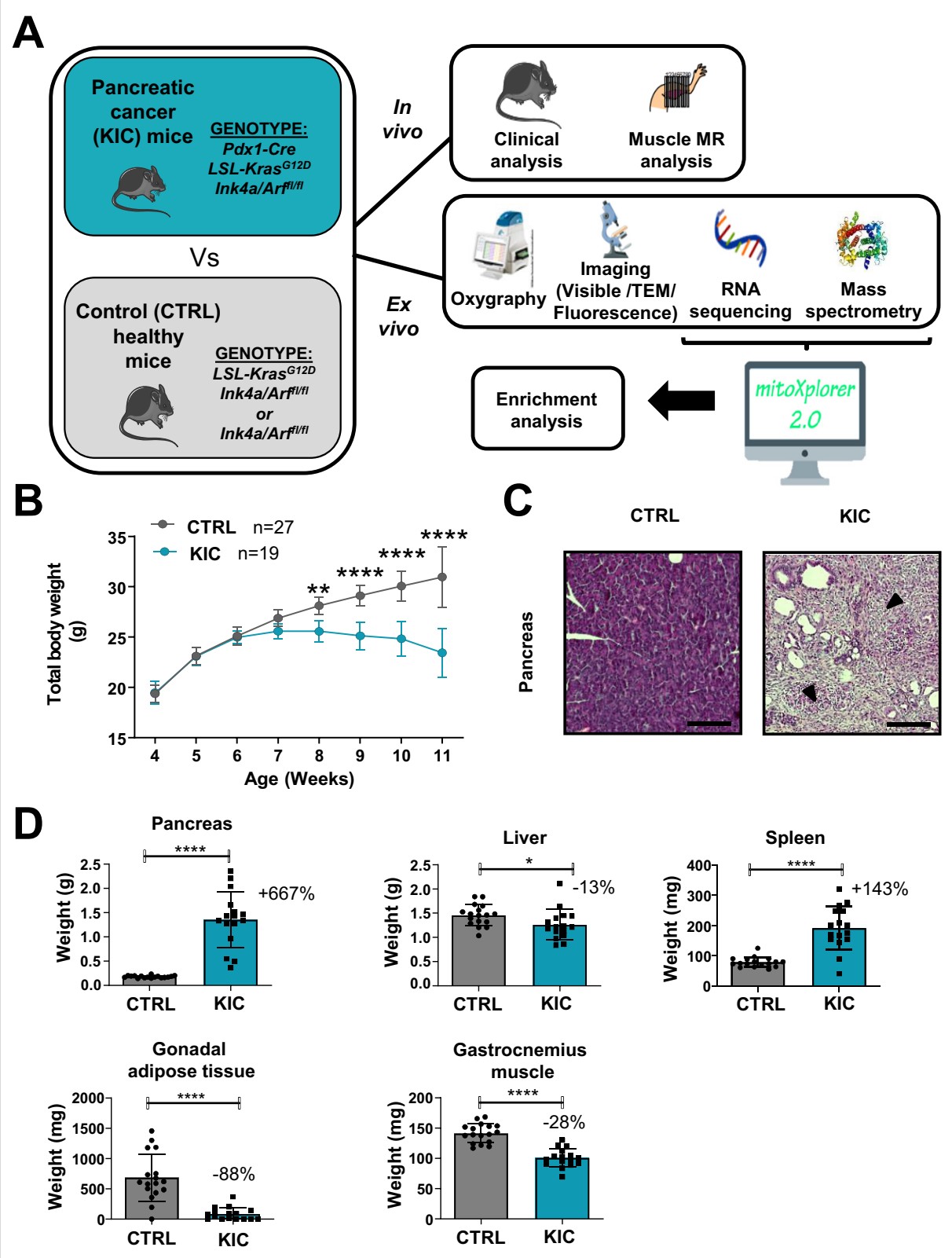

**Figure 1.** Spontaneous pancreatic ductal adenocarcinoma (PDAC) KIC mouse model shows an obvious cachexia phenotype with adipose tissue and muscle loss associated with cancer. (**A**) Schematic representation of the overall study. (**B**) Total body weight analysis during disease progression (from 4 to 11 weeks of age) in KIC and CTRL male mice. Data are mean ± 95% of CI and n represents the total number of mice analyzed in each group. Two-way ANOVA Bonferroni test; n=15 or more for control and KIC mice per time-point; **p<0.01, ****p<0.0001. (**C**) Representative hematoxylin, phloxine, and

*Figure 1 continued on next page*

*Figure 1 continued*

saffron (HPS) staining of pancreas from healthy control versus cancer KIC mice. Dark arrows point to stromal fibrotic areas relative to PDAC. Scale bar = 500 µm. (**D**) Total tissue weights of pancreas, liver, spleen, gonadal adipose tissue, and gastrocnemius muscle of healthy control mice versus end-stage (9–11 weeks) cancer KIC mice. n=17 control and n=16 KIC. Data are mean ± SD. For the gastrocnemius muscle, the weight mean of the two muscles of each individual was used to calculate the mean. Unpaired two-tailed Mann-Whitney t-tests; *p<0.05, ****p<0.0001, and n represents the number of mice that were analyzed.

stress in response to exercise activity, was similar in both phenotypes at 9–11 weeks of age. These data suggest that the tumor-induced mitochondrial defect would not impair energy supply in exercising muscle, through the compensation of the potent limitation of oxidative ATP production in 9- to 11-week-old KIC mice by another metabolic pathway. In this sense, we observed that the pH value at the end of the 6 min exercise was lower in KIC mice bearing a tumor compared to age-matched controls, thus suggesting that the tumor accelerates the glycolytic flux. Collectively, these data show a loss of gastrocnemius strength and lower ATP production by oxidative capacities in exercising KIC PDAC muscle in vivo, indicating defective mitochondria in the skeletal muscle of PDAC-bearing mice.

## Mitochondrial function and homeostasis are profoundly affected in KIC muscles

Decrease of oxidative activity in cancer KIC muscle shown in *Figure 3* (muscle from 9- to 11-week-old mice bearing a tumor) was further investigated by Seahorse oxygraphy on isolated mitochondria. Two different assays were implemented to assess coupling activity (*Figure 4A*) and respiratory complexes activity (*Figure 4B*). Quantification of mitochondrial parameters showed a decrease in basal respiration, proton leak, and maximal respiration in mitochondria from KIC muscles compared to controls, without noticeable difference in mitochondria-linked ATP production and spare respiratory capacity (*Figure 4C*). The activities of mitochondrial complexes I, II, and IV were found to be decreased (*Figure 4D*). Abundance of each respiratory complex measured by western blotting using a cocktail of oxidative phosphorylation (OXPHOS) antibodies showed decreased level of the corresponding OXPHOS proteins (*Figure 4E*). A prominent decrease of signal was observed by immunohistofluorescence microscopy using the same OXPHOS antibodies cocktail (*Figure 4F*). These data show a marked defect in mitochondrial respiratory chain in KIC PDAC muscles.

We then analyzed the mitochondrial structure by transmission electron microscopy (TEM) on longitudinal and transversal sections of the gastrocnemius muscle (*Figure 5A*). We observed striking modifications of the structure of KIC mitochondria compared to controls, with more elongated mitochondria with disorganized cristae, characterized by higher mean area and diameter and lower circularity (*Figure 5B* and *Figure 5—figure supplement 1A–C*). KIC mitochondria seemed to be hyperfused, suggesting defects in mitochondrial dynamics (fusion/fission balance) and quality control by mitophagy. Nonetheless, the number of mitochondria per surface of muscle was found unchanged (*Figure 5C*). To further address the question of mitochondrial mass, we quantified both the total quantity of mitochondria isolated from whole muscle by protein measurements and the level of mitochondrial DNA (mtDNA) in whole muscle by qPCR. We found a remarkable decrease of mitochondrial proteins and DNA levels in KIC muscles compared to control (*Figure 5D and E* and *Figure 5—figure supplement 1D and E*), which outweighed the decrease in total muscle mass (shown in *Figure 1D* and *Figure 5—figure supplement 1D*), suggesting defective mitochondrial homeostasis in sarcopenic muscle. Finally, we monitored some of the main proteins involved in mitochondrial homeostasis by western blotting (*Figure 5F*). We observed an increase in Mitofusins and OPA1 involved in mitochondrial fusion, potentially related to the elongated phenotype. Conversely, we found a decrease in Peroxisome Proliferator-activated Receptor-γ Coactivator 1 alpha (PGC1alpha) involved in mitochondrial biogenesis and in Transcription Factor A Mitochondrial (TFAM) involved in mtDNA replication, repair, and transcription. These data demonstrate strong mitochondrial qualitative and quantitative defects in PDAC gastrocnemius muscle.

## Molecular pathways involved in mitochondrial homeostasis, metabolism, and function are strongly dysregulated in PDAC muscle

To go deeper into the molecular mechanisms underlying mitochondrial defects in PDAC muscles, we performed large-scale transcriptomic and proteomic analyses. Overall, a wealth of information was

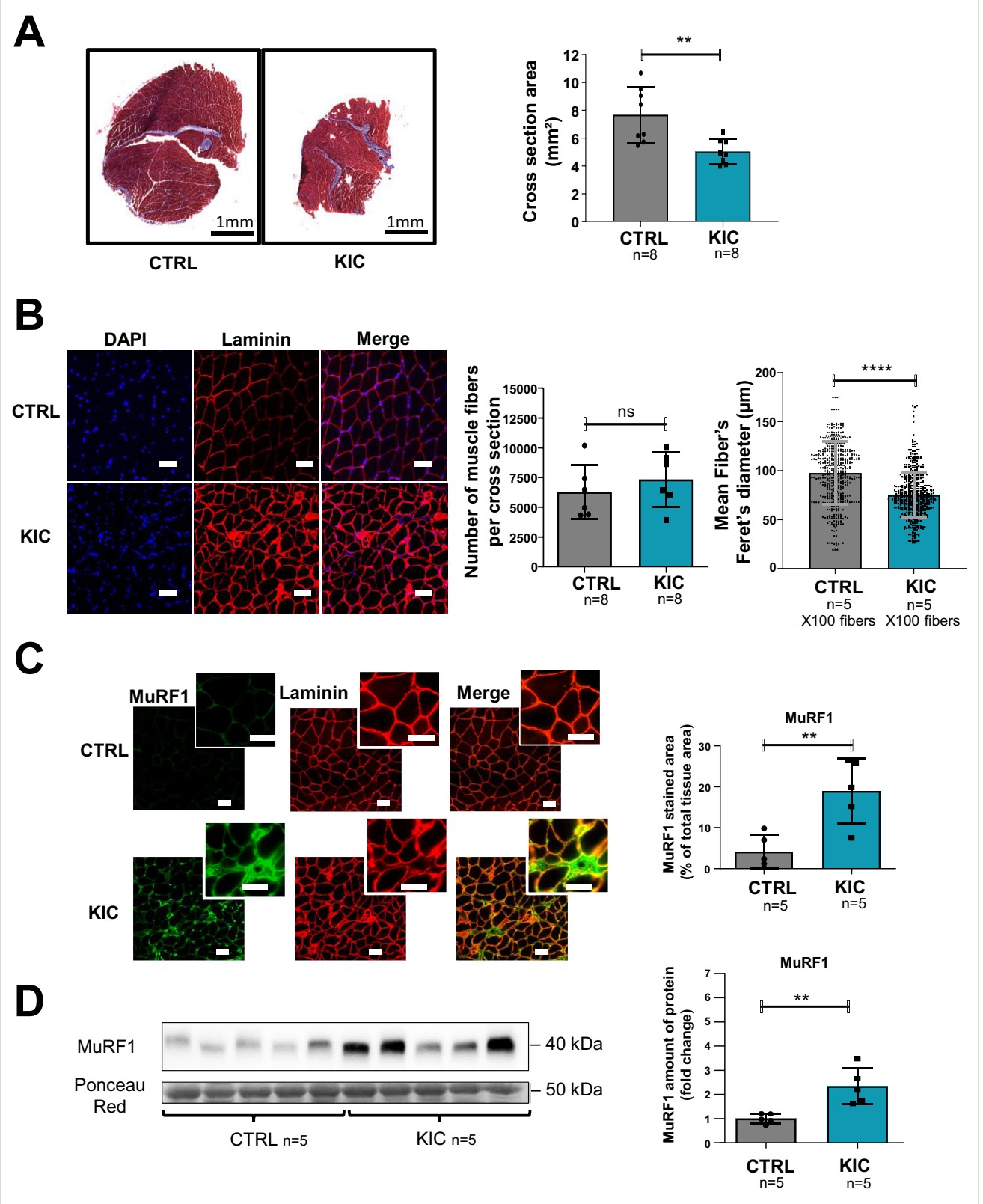

**Figure 2.** Pancreatic ductal adenocarcinoma (PDAC) mice show muscle atrophy associated with fiber's size decrease and activation of the proteasome pathway. (**A**) (Left) Representative images of transversal sections of gastrocnemius muscle from cancer KIC male mice (9- to 11-week-old bearing a tumor) and age-matched healthy control, following Masson's trichrome staining. Scale bars, 1 mm. (Right) Diagrams of cross-sectional areas of gastrocnemius muscle from healthy control mice and cancer KIC mice (n=8/group). Data are mean ± SD. Unpaired two-tailed Mann-Whitney t-tests;

*Figure 2 continued on next page*

*Figure 2 continued*

**p<0.01, and n represents the number of mice that were analyzed. (**B**) (Left) Representative immunostaining of gastrocnemius muscle cross sections from control and KIC mice stained with DAPI (blue; nuclei), and anti-laminin antibody (red) to evaluate muscle fiber's size. Scale bars, 40 μm. (Right) Number of muscle fibers counted in gastrocnemius cross sections from control and KIC mice, and mean fiber's (100) minimal Feret's diameter in gastrocnemius muscle from control and KIC mice. Data are mean ± SD. Unpaired two-tailed Mann-Whitney t-tests; ns = non-significant, ****p<0.0001, and n represents the number of mice that were analyzed. (**C**) (Left) Representative immunostaining of transverse sections of gastrocnemius muscle from control and KIC male mice stained for anti-MuRF1 (green) to evaluate muscle atrophy, anti-laminin (red) to evaluate muscle fibers, and merged images. Scale bars, 40 μm. (Right) MuRF1 immunofluorescent staining quantification in transversal sections of gastrocnemius muscle from control and KIC mice. Data are mean ± SEM. Unpaired one-tailed Mann-Whitney t-tests; **p<0.01 and n represents the number of mice that were analyzed. (**D**) (Left) Representative immunoblot for MuRF1 in gastrocnemius muscle from CTRL and KIC mice (n=5/group). The loading control indicated is the Ponceau Red staining of the membrane at the 50 kDa level (the same membrane as in *Figure 4E* is shown in duplicate). (Right) Quantification of immunoblots to show the MuRF1 protein levels in gastrocnemius muscle from CTRL and KIC mice. The relative protein amount was calculated from the control mean in two different experiments. n=5 mice/group. Data are mean ± SD. Unpaired two-tailed Mann-Whitney t-tests; **p<0.01 and n represents the number of mice that were analyzed.

The online version of this article includes the following source data and figure supplement(s) for figure 2:

**Source data 1.** Uncropped and labeled gels for *Figure 2*.

**Source data 2.** Raw unedited gels for *Figure 2*.

**Figure supplement 1.** Decreased fiber's size but moderate change in proportion of the different fiber types in pancreatic ductal adenocarcinoma (PDAC) atrophic muscle (related to *Figure 2*).

---

obtained from these two approaches, such that we try to summarize the major trends below. Please note that we have mainly used the web tool Ingenuity Pathways Analysis (IPA) for analyses, and that we display all differential pathways regardless of their z-score (not just the 2 < z-score < –2) and directionality, as we believe they are all interesting to consider.

## Transcriptomic analysis

RNA sequencing was done on RNA extracted from gastrocnemius muscle from 9- to 11-week-old KIC mice bearing a tumor and age-matched controls. A total of 2304 genes were found to be dysregulated in KIC compared to control muscles, with 1277 upregulated and 1027 downregulated (*Figure 6—figure supplement 1A*; up- and downregulation in blue and red color, respectively). The web-based IPA shown in *Figure 6—figure supplement 2* pointed to pathways which were expected to be dysregulated, such as autophagy, catabolism of protein, glycolysis, and atrophy of muscle with a positive z-score, and skeletal muscle physiology (cell movement of muscle cells, migration of muscle cells, function of skeletal muscle) with a negative z-score. By querying the RNA sequencing data tables, we found that the genes encoding the E3 ubiquitin ligase MuRF1 and MAFBx involved in muscle atrophy (*Trim63* and *Fbxo32*, respectively) are highly over-expressed in KIC muscles as expected (six- and fivefold increase, respectively).

We then used the data mining mitoXplorer web-based platform to focus our analysis on mitochondria-associated pathways (*Figure 6* and *Figure 6—figure supplement 1*, *Figure 6—figure supplement 3* and *Figure 6—figure supplement 4*). A total of 177 genes from the mitoXplorer list of mitochondrial interactome genes were found to be dysregulated in KIC compared to control muscles, with 85 upregulated and 92 downregulated (*Figure 6—figure supplement 1B*). IPA of mitochondrial dysregulated pathways (*Figure 6*) showed four pathways in common with the general IPA: autophagy, glycolysis, atrophy of muscle, and organization of cytoplasm. In addition, this mitochondrial IPA pointed to nine pathways involved in mitochondrial homeostasis and respiratory function, consistent with decreased functional OXPHOS in KIC muscle: depolarization, swelling, and permeabilization of mitochondria with positive z-scores, and respiration, transmembrane potential, quantity, fragmentation and density of mitochondria, and consumption of oxygen with negative z-scores. Interestingly, besides mitochondrial homeostasis, mitochondrial IPA identified three lipid metabolism pathways (accumulation of lipids, oxidation of lipid, and oxidation of long chain fatty acid), to put in link with the fatty acid metabolism pathway with a positive z-score in the general IPA. The mitoXplorer visual representation of dysregulated genes highlighted amino acid metabolism and OXPHOS as highly dysregulated with multiple up- and downregulated genes, respectively (*Figure 6—figure supplements 3 and 4*). Translation and glycolysis were also among the most dysregulated pathways. Interestingly, metabolic pathways feeding the respiratory chain were found dysregulated: half of the

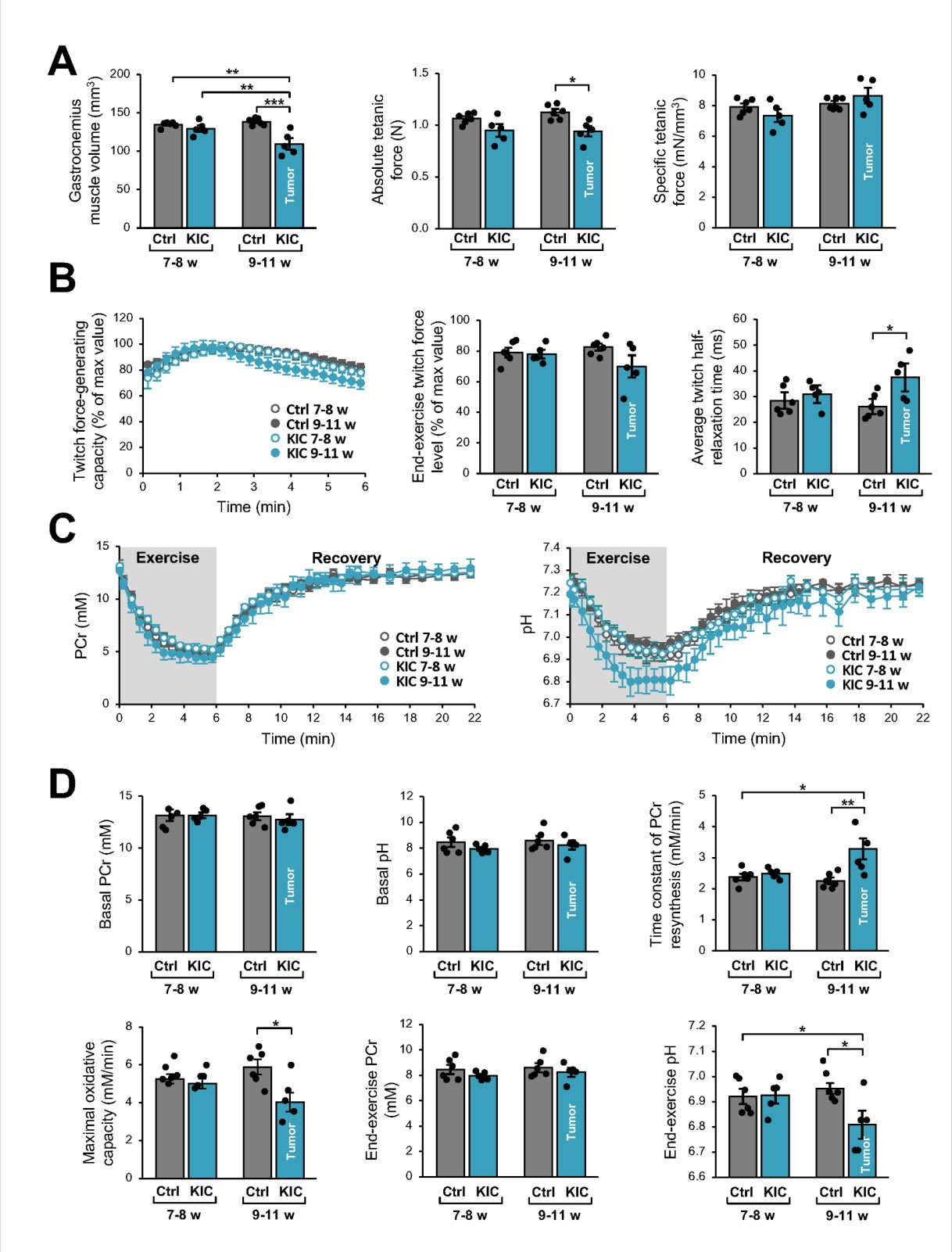

**Figure 3.** Noninvasive magnetic resonance (MR) investigation of gastrocnemius muscle function and bioenergetics. This longitudinal investigation was done in the same KIC mice, at 7- to 8-week-old (pre-tumoral) and at 9–11 weeks of age (bearing a tumor), and same age-matched control mice. (**A**) (Left) Gastrocnemius muscle volume was calculated from MR images. (Middle and right) Absolute and specific force production in response to a tetanic electrostimulation train (150 Hz; 0.5 s duration). (**B**) (Left) Twitch force-generating capacity (expressed as percent of maximal twitch force value)

*Figure 3 continued on next page*

*Figure 3 continued*

during the 6 min bout of exercise performed simultaneously to the dynamic [31]P-MRS acquisition. (Middle) Twitch force level at the end of the 6 min bout. (Right) Twitch half-relaxation time averaged over the whole bout of exercise. (**C**) Changes in phosphocreatine (PCr) level (left) and pH (right) throughout the 6 min exercise and the 16 min post-exercise recovery period; for both panels, the first time-point (t=0) indicates the basal value. (**D**) Basal PCr level and pH were measured in resting muscle. Time constant of PCr resynthesis and maximal oxidative capacity were measured during recovery. The drop of PCr and acidosis were determined at the end of the 6 min bout of exercise. Data are means ± SEM. Controls, n=7; KIC, n=5. All samples were normally distributed according to the Shapiro-Wilk test. Significant differences were determined by two-factor (group × age) repeated measures ANOVA followed when appropriate by Tukey-Kramer post hoc multiple comparison tests. *p<0.05, **p<0.01, ***p<0.001.

genes involved in the tricarboxylic acid (TCA) cycle were downregulated, and four lipid metabolism pathways (fatty acid degradation and beta-oxidation, fatty acid biosynthesis and elongation, fatty acid metabolism, and metabolism of lipids and lipoproteins) were affected. Moreover, this analysis highlighted dysregulations of mitochondrial dynamics and mitophagy genes, in accordance with mitochondrial structural alterations observed by TEM.

Notably, the genes encoding PGC1 proteins - *Ppargc1a* encoding PGC1α and *Ppargc1b* encoding PGC1ß - involved in mitochondrial biogenesis were strongly down-expressed (see the 'Transcription (nuclear)' heatmap in *Figure 6—figure supplement 4*). Finally, genes involved in ROS defenses were found to be upregulated ('ROS defense' group *Figure 6—figure supplement 4*), including *Nfe2l2* encoding the Nrf2 antioxidant transcription factor (in the 'Replication & Transcription' group *Figure 6—figure supplement 4*), which is a mark of oxidative stress.

## Proteomic analysis

We also did a proteomic analysis by mass spectrometry of proteins extracted from entire muscle. A total of 154 proteins were found to be dysregulated in KIC compared to control muscles, with 107 upregulated and 47 downregulated (*Figure 7—figure supplement 1A*). IPA data shown in *Figure 7—figure supplement 2* pointed to dysregulation of muscle physiology pathways (contractility of muscle, muscle contraction, and organization of muscle cells) and multiple metabolic pathways (mostly lipid and protein metabolic pathways, but also nucleotide and DNA metabolism). Thus, dysregulations of muscle physiology and metabolism are common between the general proteomic and transcriptomic analyses.

We then focused our proteomic analysis on mitochondria-associated pathways using the mitoXplorer platform (*Figure 7*, *Figure 7—figure supplement 1B and 4*, *Figure 7—figure supplements 3 and 4*). IPA of mitochondrial dysregulated pathways identified two pathways (synthesis of ROS and oxidative stress) that are connected to two pathways found in the general IPA (metabolism of hydrogen peroxide and quantity of ROS), all related to redox metabolism and with positive z-scores. In addition, the mitochondrial IPA highlighted five pathways involved in mitochondrial energy and apoptotic functions: consumption of oxygen and apoptosis (with positive z-scores), and concentration of ATP, synthesis of ATP, and cell viability (with negative z-scores). The mitoXplorer visual representation of dysregulated proteins pointed to OXPHOS, translation, amino acid metabolism and TCA cycle as the major dysregulated pathways (*Figure 7—figure supplement 3*), which is consistent with the transcriptomic data. Surprisingly, the ROS defense proteins were all found downregulated at the protein level, when compared with upregulated ROS defense genes in the transcriptomic data, which could suggest an altered response to major oxidative stress in KIC muscles.

## Common mitochondria-associated pathways between transcriptomic and proteomic analyses

To compare the outcome of transcriptomic and proteomic IPA analyses regarding mitochondrial dysregulated pathways in cachectic cancer muscle, we first highlight the shared (common) pathways in *Figure 7*; there are three common pathways with the same z-score directionality (production/synthesis of ROS, glycolysis, and metabolism of amino acids) and four with opposite z-score directionality (DNA damage, oxidative stress, consumption of oxygen, and autophagy). Second, we show heatmaps of the genes or proteins belonging to each of these seven common pathways (*Figure 7—figure supplement 5*; *Figure 7—figure supplement 6* and *Figure 7—figure supplement 7*); the *Dlst* gene, encoding the TCA cycle dihydrolipoyllysine-residue succinyltransferase (Dlst) protein, was the only one to be found in both transcriptomic and proteomic signatures (production/synthesis of ROS),

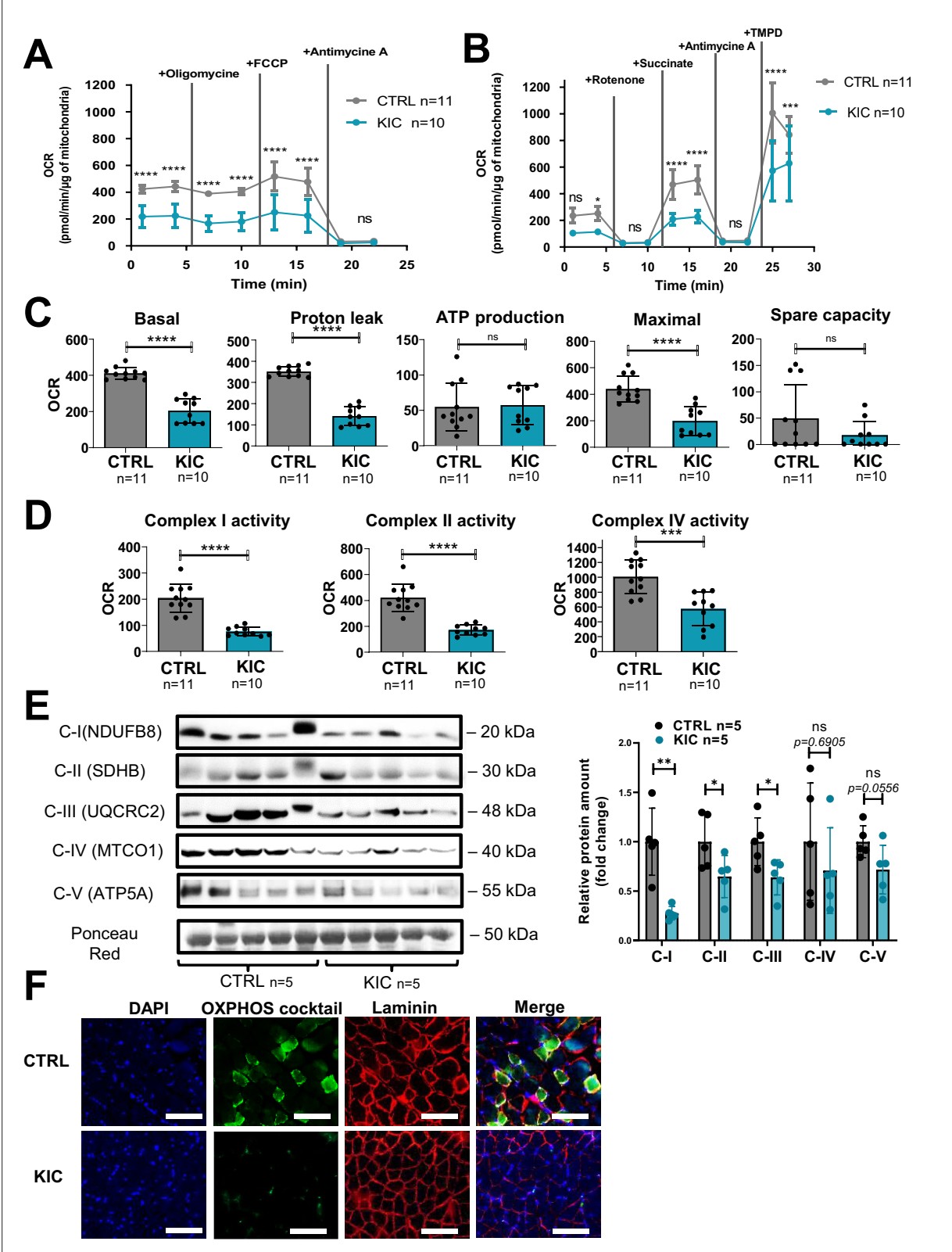

**Figure 4.** Cancer-associated sarcopenia is correlated with a decrease of mitochondrial respiratory activity and respiratory complexes. (**A,** **B**) Representative Seahorse XF oxygen consumption rate assays performed on mitochondria isolated from gastrocnemius muscle of age-matched healthy control and cancer (9- to 11-week-old bearing a tumor) KIC mice. The curves were normalized to 1 µg of mitochondrial protein (the assays were done with 2.5 µg of mitochondria). (**A**) Mitochondrial coupling assay. The analysis medium was supplemented with ADP (5 mM), succinate (10 mM), and

*Figure 4 continued on next page*

*Figure 4 continued*

rotenone (2 µM). (**B**) Electron transport chain assay performed on the same samples of isolated mitochondria. The analysis medium was supplemented with pyruvate (5 mM), malate (1 mM), and FCCP (4 µM) to allow maximal respiration. Data are mean ± SD and n represents the number of mice analyzed. Two-way ANOVA Bonferroni test; ns = non-significant, *p<0.05, ***p<0.001, ****p<0.0001. (**C, D**) Quantification of the different parameters of mitochondrial respiratory activity, normalized per µg of mitochondrial protein. (**C**) Parameters calculated from the mitochondrial coupling assay. From left to right: basal respiration, measured as the basal OCR subtracted from the background OCR (i.e. OCR after antimycin A addition); proton leak, measured as the OCR after oligomycin addition subtracted from the background OCR; ATP production from mitochondria, measured as the basal OCR subtracted from the OCR after oligomycin addition; maximal respiration, measured as the OCR after FCCP addition subtracted from the background OCR; spare capacity, i.e., the difference between maximal and basal mitochondrial respiration. (**D**) Respiratory complexes activities calculated from the OCR values obtained in the electron transport chain assay. From left to right: complex I activity, measured as the basal OCR subtracted from the OCR after rotenone addition; complex II activity, measured as the difference between succinate-driven OCR and OCR after antimycin A addition; complex IV respiration, measured as the difference between TMPD-driven OCR and OCR after antimycin A addition. Data are mean ± SD. Unpaired two-tailed Mann-Whitney t-tests; ns = non-significant, ***p<0.001, ****p<0.0001; n represents the number of mice analyzed per condition. (**E**) Left: Representative immunoblot for mitochondrial complexes using the rodent cocktail oxidative phosphorylation (OXPHOS) antibodies specific of one protein for each complex: CI (NDUFB8), CII (SDHB), CIII (UQCRC2), CIV (MTCO1), CV (ATP5A), in gastrocnemius muscle from CTRL and KIC mice (n=5/group). The loading control indicated is the Ponceau Red staining of the membrane at the 50 kDa level. Right: Relative protein amount of CI (NDUFB8), CII (SDHB), CIII (UQCRC2), CIV (MTCO1), CV (ATP5A) in gastrocnemius muscle from CTRL and KIC male mice. The relative protein amount was calculated from the mean of CTRL from two different experiments. n=5 mice/group. Data are mean ± SD. Unpaired two-tailed Mann-Whitney t-tests; ns = non-significant with p-value indicated above, *p<0.05, **p<0.01. (**F**) Representative immunostaining of gastrocnemius muscle cross sections from control (n=5) and KIC (n=5) mice stained for mitochondrial electron transport chain complexes I to V (green) to evaluate mitochondrial mass, DAPI (blue) to evaluate nucleus, laminin (red) to evaluate muscle fibers, and merged images as indicated. Scale bars, 100 µm.

The online version of this article includes the following source data for figure 4:

**Source data 1.** Uncropped and labeled gels for *Figure 4*.

**Source data 2.** Raw unedited gels for *Figure 4*.

and it is downregulated for both gene expression and protein level. Third, comparison of mitoXplorer transcriptomics and proteomics heatmaps (*Figure 6—figure supplement 4* and *Figure 7—figure supplement 4*) shows 16 common genes/proteins, among them 8 show same regulation orientation at the level of gene expression and protein, and 8 show opposite (*Table 1*). Collectively, these data highlight many mitochondrial genes and their encoded proteins whose expression and/or stability are dysregulated in cancer cachectic muscle compared to healthy muscle. Several of these genes/proteins are involved in biological processes that are still poorly explored in cancer-associated cachexia, such as nucleotide and folate metabolism, which is of interest in the context of mtDNA homeostasis.

## Discussion

This study provides an in-depth deciphering of mitochondrial dysfunction during PDAC-induced muscle atrophy at the tissular, cellular, and molecular level. Cachexia in tumor-bearing KIC mice is characterized by almost complete disappearance of the gonadal adipose tissue and loss of mass (weight and volume) and strength in gastrocnemius muscle, consistent with the previously reported tissue wasting in PDAC mouse models (*Danai et al., 2018*; *Michaelis et al., 2017*). Muscle mass loss is associated with a diminution in muscle fiber's size, which is a common feature in cancer muscle wasting studies (*Johns et al., 2013*). In contrast, no significant changes in the number and type of fibers were observed, in agreement with the fact that not all studies have demonstrated type 1 and 2 fiber differences in cancer muscle wasting studies (*Johns et al., 2013*). Fibers' size reduction is due to decreased protein synthesis and increased proteolysis, the latter involving the proteasome and autophagy protein degradation processes. Accordingly, we evidenced induction in KIC muscle of the E3 ubiquitin ligase MuRF1 (RNA and protein) and MAFBx (RNA) involved in the proteasomal protein degradation (*Argilés et al., 2014*). We also found upregulation of the *Sqstm1* gene encoding p62 involved in autophagy, as expected (*VanderVeen et al., 2017*). The detailed analysis of gastrocnemius muscle bioenergetics by noninvasive multimodal MR demonstrates lower ATP production by oxidative capacities in KIC PDAC mice that is at least partly compensated by increased glycolysis, which could account for apparently normal muscle activity during exercise. MR data point to dysfunctional mitochondria in the skeletal muscle of PDAC-bearing mice, which was confirmed by oxygraphy (Seahorse) experiments performed on isolated mitochondria, showing a large reduction of mitochondrial respiration in KIC muscle likely as the result of decreased respiratory complexes activity and abundance.

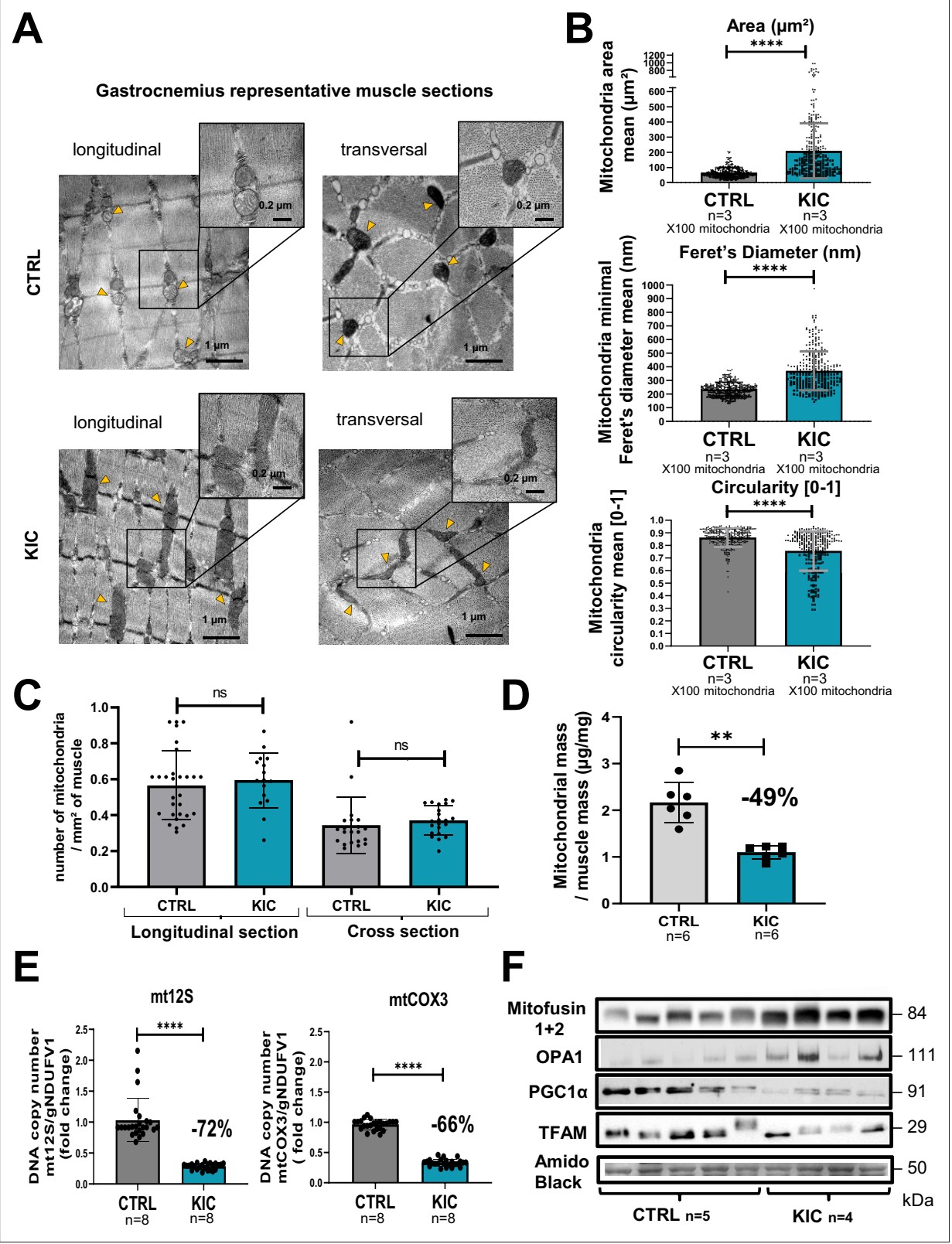

**Figure 5.** Mitochondrial structure is profoundly altered in cancer KIC muscles. (A–C) Transmission electron microscopy (TEM) analysis of muscles showing altered mitochondria in cancer mouse muscle. (A) Representative TEM micrographs of longitudinal and transversal sections of gastrocnemius muscles from CTRL and cancer KIC mice showing mitochondrial ultrastructure. Scale bar = 1 or 0.2 μm. Yellow triangles point to mitochondria. (B) Quantification of mitochondrial morphological characteristics. From top to bottom: Mitochondrial area (μm²), mitochondrial minimal Feret's diameter,

*Figure 5 continued on next page*

*Figure 5 continued*

and mitochondrial circularity in longitudinal sections of gastrocnemius muscles from CTRL healthy mice and cancer KIC mice. n=3 mice/group and 100 mitochondria measured/mouse. Data are mean ± SD. Unpaired two-tailed Mann-Whitney t-test. **** p<0.0001. (**C**) Quantification of the number of mitochondria per mm² in longitudinal and transversal sections of gastrocnemius muscles from CTRL healthy mice and cancer KIC mice. n=3 mice/group and 15 or more pictures have been taken and measured/mouse. Data are mean ± SD. Unpaired two-tailed Mann-Whitney t-test; ns = non-significant. (**D**) Mitochondrial mass of total isolated mitochondria (in µg of proteins), normalized by the related entire gastrocnemius muscle mass (in mg) in CTRL and KIC mice. Data are mean ± SD; n=6 mice/group. Unpaired two-tailed Mann-Whitney t-tests; **p<0.01. (**E**) Relative mitochondrial DNA content in gastrocnemius muscle of CTRL and KIC mice using relative level of *mtCOX3* (left) and *mt12S* (right) as mitochondrial genes normalized on *gNDUFV1* as nuclear gene. Data are mean ± SD, n=8 mice/group. Unpaired two-tailed Mann-Whitney t-tests, ****p<0.0001. (**F**) Representative immunoblots for Mitofusin1+2 and OPA1 (involved in mitochondrial fusion), PGC1α (involved in mitochondrial biogenesis), and TFAM (involved in mitochondrial DNA replication, repair, and transcription) in gastrocnemius muscle from CTRL (n=5) and KIC (n=4) mice. The loading control indicated is the Amido Black staining of the PGC1α membrane at the 50 kDa level.

The online version of this article includes the following source data and figure supplement(s) for figure 5:

**Source data 1.** Uncropped and labeled gels for *Figure 5*.

**Source data 2.** Raw unedited gels for *Figure 5*.

**Figure supplement 1.** Mitochondrial morphology is profoundly altered in KIC muscles (related to *Figure 5*).

Finally, we observed important modifications in the size and shape of mitochondria in PDAC muscle with the presence of elongated mitochondria in accordance with the relation between mitochondrial function and structure (*Bulthuis et al., 2019*), and decreased mitochondrial mass without any change in the total number of mitochondria. Collectively, these data demonstrate profound quantitative and qualitative alterations of mitochondria in pancreatic cancer muscle.

The major originality of this study is the deep deciphering of mitochondrial molecular changes at the transcriptomic and proteomic level, providing a large quantity of data that improve the knowledge of mitochondrial dysfunction in PDAC cachexia.

As expected, metabolism of amino acids and glycolysis were among the most dysregulated pathways in PDAC cachectic muscle, consistent with the overall altered protein metabolism and loss of OXPHOS function, respectively. Redox metabolism was also found to be altered, which has to be related to deregulation of the inflammatory response shown in *Figure 7—figure supplement 4* (*VanderVeen et al., 2017*). Increased expression of multiple ROS defense genes is consistent with oxidative stress prone to affect mitochondrial macromolecules and homeostasis (*Penna et al., 2020*; *Zhang et al., 2023*). It should be noted that the proteins involved in antioxidant defense are, on the contrary, downregulated, suggesting their defective production to fight against oxidative stress. Collectively, these data strongly support that the main molecular alterations in KIC mitochondria are related to energy and redox metabolisms; importantly, these metabolisms are closely linked in a vicious circle, with ROS damaging mitochondria and, in turn, dysfunctional OXPHOS promoting oxidative stress (*Penna et al., 2020*; *Zhang et al., 2023*).

Regarding mitochondrial dynamics, we looked at proteins involved in mitochondrial fusion (Mitofusins 1 and 2 and OPA1) and fission (DRP1 and FIS1). We found a slight decrease of both *Opa1* and *Drp1* gene expression in KIC muscle (no difference for *Mfn1*, *Mfn2*, and *Fis1*). In contrast, we found the proteins OPA1 and Mitofusin1+2 as upregulated in KIC muscle by western blotting, suggesting that transcriptomic and proteomic modifications of molecules involved in mitochondrial fusion are not related. Regarding mitophagy, the genes *Bnip3*, *Bnip3l*, and *Pink1* were found downregulated in KIC muscle. One could imagine the occurrence of mitochondrial hyperfusion in the KIC context as a compensatory mechanism allowing to maintain (rather badly than well…) the functioning of mitochondria in this muscle in full catabolism, and also perhaps to preserve the mitochondria from degradation by autophagy (*Gomes et al., 2011*). Finally, the genes encoding PGC1α and PGC1β involved among other roles in mitochondrial biogenesis were strongly down-expressed as expected (*Penna et al., 2020*; *VanderVeen et al., 2017*).

Major novelties revealed by this study concern the lipid and nucleotide metabolisms. Lipid metabolism pathways were found to be downregulated in KIC muscles and mitochondria, which has to be related to the stringent loss of fat mass in cachectic KIC mice and the physiological role of fatty acid in mitochondrial respiration via fatty acid oxidation (FAO). Interestingly, it was reported that excessive FAO induces muscle atrophy in cancer cachexia (*Fukawa et al., 2016*), suggesting that early deregulation of mitochondrial lipid metabolism could be involved in triggering muscle atrophy in

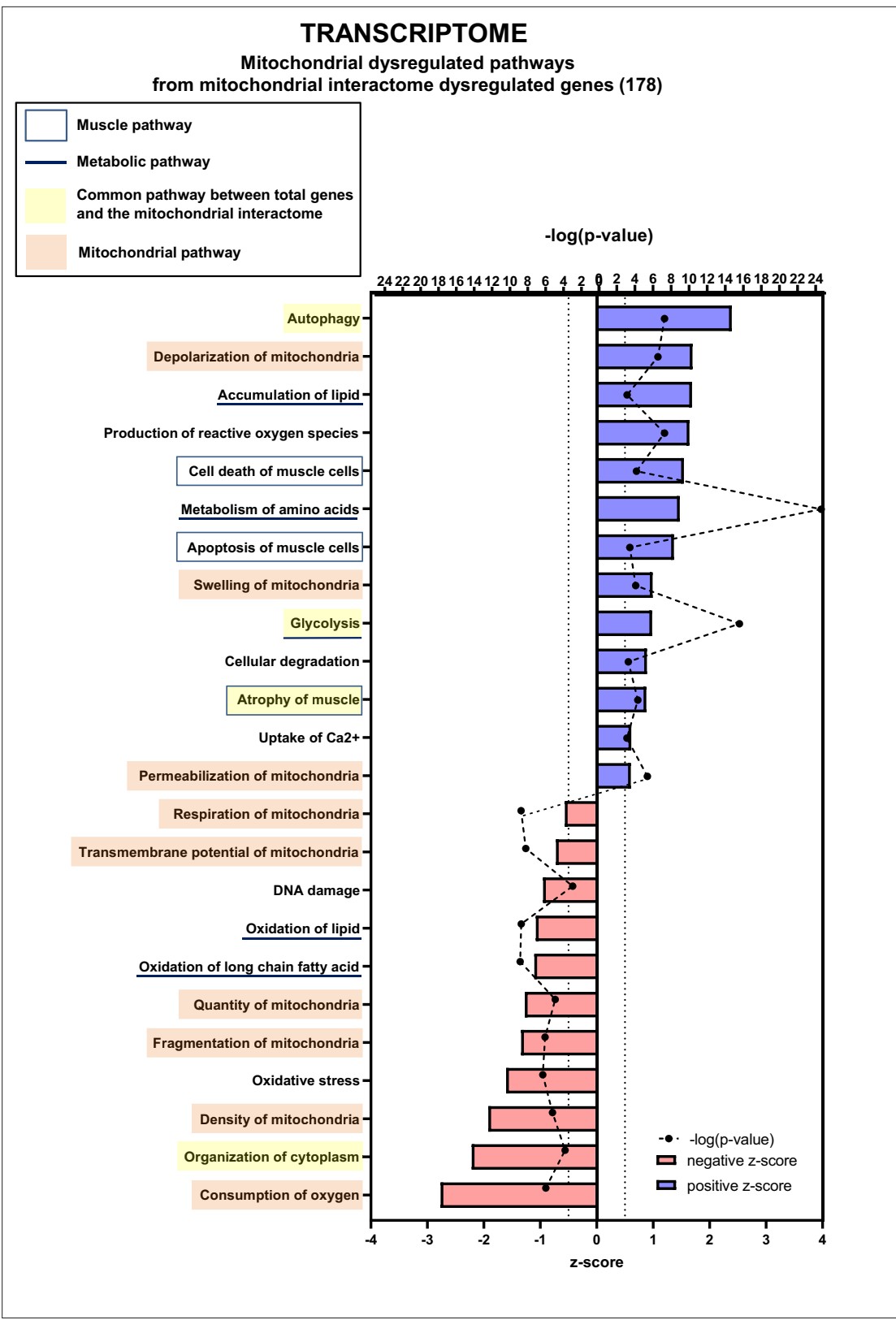

**Figure 6.** Transcriptomic analysis demonstrates multiple mitochondria-associated gene dysregulations in cancer sarcopenic muscles. Ingenuity Pathways Analysis (IPA) showing functions, pathways, and diseases significantly dysregulated in gastrocnemius muscle of cancer KIC mice compared to healthy control mice, associated with genes of the mitochondrial interactome extracted from mitoXplorer tool with significant dysregulations (false discovery rate [FDR]>0.05)=178 genes. Each process is represented using z-score>0.5 as upregulated (blue) or downregulated (red), and the significance

*Figure 6 continued on next page*

*Figure 6 continued*

represented as –log(p-value) with black dots connected with a dotted line. Muscle pathways are framed by a square. Common pathways between total genes and the mitochondrial interactome are highlighted in yellow. Mitochondrial pathways are highlighted in orange. Other metabolic pathways are underlined.

The online version of this article includes the following figure supplement(s) for figure 6:

**Figure supplement 1.** Transcriptomic analysis of cancer sarcopenic muscles (related to *Figure 6*).

**Figure supplement 2.** Transcriptomic analysis of total dysregulated genes in cancer sarcopenic muscles (related to *Figure 6*).

**Figure supplement 3.** Mitochondrial interactome of down- or upregulated genes involved in mitochondrial processes in KIC gastrocnemius muscle compared to CTRL (muscle RNA n=4 for each condition).

**Figure supplement 4.** Cancer sarcopenic muscles demonstrate obvious mitochondria-associated genes dysregulation (related to *Figure 6* and *Figure 6—figure supplement 3*).

PDAC. This remains to be investigated. In addition, several pathways of DNA metabolism, such as DNA damage, degradation of DNA, nucleotide synthesis, and folate pathway (involved in purines and pyrimidines synthesis), were found deregulated, suggesting a certain level of DNA instability affecting the mitochondrial genome. Interestingly, the decrease of mtDNA in KIC muscle was even larger than the decrease in mitochondrial mass as measured by protein content, suggesting altered mitochondrial genome homeostasis in PDAC-induced sarcopenia, as has been reported in age-related sarcopenia (*Barbieri et al., 2015*; *Rygiel et al., 2016*). It would be interesting to further investigate the interactions between mtDNA alterations and redox and nucleotide metabolism, in the context of a growing understanding of the importance of ROS and mtDNA released during the inflammatory response (*Barnett et al., 2023*). In sum, the large amount of data generated by this study provides a wealth of information for the identification of new gene and protein markers of cachexia associated with pancreatic cancer.

A transcriptomic analysis of PDAC cachectic muscle was reported (*van der Ende et al., 2018*) using the gene expression microarrays data generated on biceps femoris muscle from KIC mice (*Gilabert et al., 2014*). It highlighted decreased expression of genes involved in mitochondrial fusion, fission, ATP production, and mitochondrial density, and increased expression of genes involved in ROS detoxification and mitophagy. We observed these same gene modulations in the same direction in KIC gastrocnemius. In a notable way, we have here largely completed this preliminary study by integrating transcriptomic and proteomic data in cachectic muscle, and by revealing a large number of modulated mitochondrial genes. Importantly, integrating the transcriptome and proteome in KIC muscle showed several common pathways that are deregulated during cachexia, suggesting that considering transcriptome only (for which diagnostic tools are available in the clinic) would be relevant in further studies. Importantly, in major pathways, deregulation of genes in one direction (up or down) can be the opposite for the corresponding protein (down or up, respectively). This is consistent with the fact that for one gene, mRNA level (related to gene transcription rate and RNA stability) is not obligatory correlated with protein level (related to translation rate and protein stability) (*Liu et al., 2016*; *Piccirillo et al., 2014*).

The first limitation of this study is the use the KIC PDAC model which is fast (9–11 weeks) compared to human pathology. Nevertheless, this model is indicative of common molecular pathways involved in cachexia in human and mouse; for example, disruption of mitochondrial morphology and decrease of mtDNA were also observed in the skeletal muscle of cachectic patients (*de Castro et al., 2019*). Second, we used only male mice, yet some mechanisms of pancreatic cancer cachexia are sex specific (*Zhong et al., 2022*). We postulated that mitochondrial alterations would be similar in males and females, but this question remains to be addressed. Third, this study is lacking molecular data concerning mitochondria in pre-cachectic KIC muscle before the tumor-induced cachexia, that we monitored with noninvasive multimodal MR measurements (7- to 8-week-old [pre-tumoral] KIC mice in *Figure 3*). It will be very interesting to know whether mitochondrial defects happen before the start of muscle atrophy in PDAC, and initiate sarcopenia as demonstrated in other studies (*Alway et al., 2017*; *Brown et al., 2017*; *Waltz et al., 2018*).

In conclusion, this work demonstrates that muscle atrophy is associated with strong mitochondrial metabolic defects that are not limited to carbohydrates and proteins, but also concern lipids and nucleic acids. It presents important translational perspectives in the clinic considering that

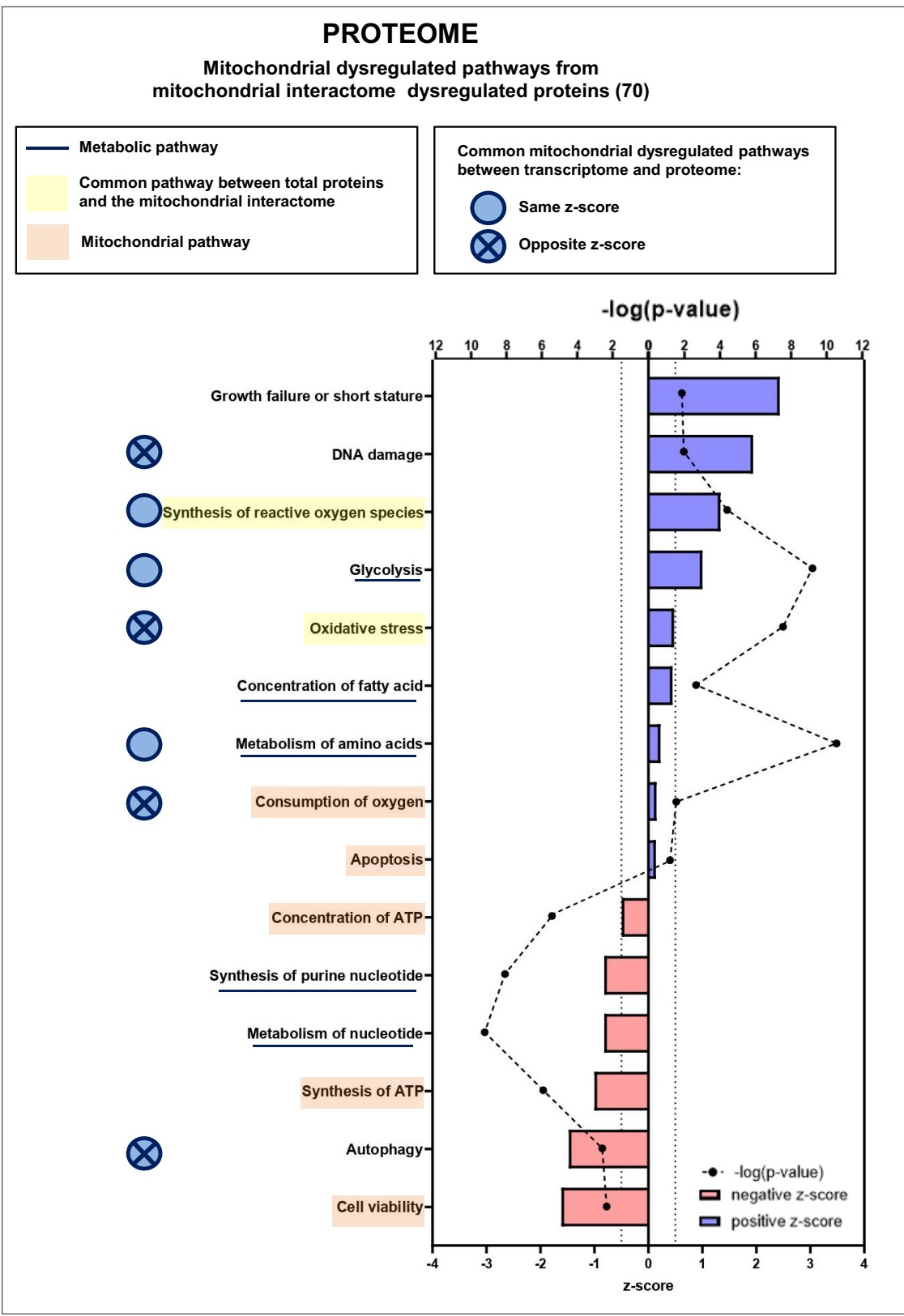

**Figure 7.** Proteomic analysis shows that numerous mitochondrial proteins are dysregulated in cancer sarcopenic muscle. Ingenuity Pathways Analysis (IPA) enrichment analysis of dysregulated functions, pathways, and diseases in gastrocnemius muscle of cancer KIC compared to control mice, associated with proteins of the mitochondrial interactome extracted from mitoXplorer tool with significant dysregulations (p-value<0.05, peptide>1, log2(fold change)>0.5)=53 proteins. Each process is represented using z-score>0.5 as upregulated (blue) or downregulated (red), and the significance

*Figure 7 continued on next page*

*Figure 7 continued*

represented as –log(p-value) with black dots connected with a dotted line. Common pathways between total proteins and the mitochondrial interactome are highlighted in orange. Mitochondrial pathways are highlighted in yellow. Muscle pathways are framed by a square. Common pathways between total genes and the mitochondrial interactome are highlighted in yellow. Mitochondrial pathways are highlighted in orange. Other metabolic pathways are underlined. Common mitochondrial dysregulated pathways between transcriptome and proteome are indicated by a blue circle which is empty when the z-score is the same and contains a cross when the z-score is opposite.

The online version of this article includes the following figure supplement(s) for figure 7:

**Figure supplement 1.** Proteomic analysis of cancer sarcopenic muscles (related to *Figure 7*).

**Figure supplement 2.** Proteomic analysis showing total proteins dysregulated in cancer sarcopenic muscle (related to *Figure 7*).

**Figure supplement 3.** Mitochondrial interactome of down- or upregulated proteins involved in mitochondrial processes in KIC gastrocnemius muscle compared to CTRL.

**Figure supplement 4.** Cancer sarcopenic muscles demonstrate obvious mitochondria-associated protein dysregulation (related to *Figure 7* and *Figure 7—figure supplement 3*).

**Figure supplement 5.** Common mitochondrial dysregulated pathways between transcriptomic and proteomic analyses (related to *Figures 6 and 7*).

**Figure supplement 6.** Common mitochondrial dysregulated pathways between transcriptomic and proteomic analyses (related to *Figures 6 and 7*).

**Figure supplement 7.** Common mitochondrial dysregulated pathways between transcriptomic and proteomic analyses (related to *Figures 6 and 7*).

mitochondrial dysfunction was also observed in cachectic cancer patients (*de Castro et al., 2019*; *Dolly et al., 2022*). Therefore, our data provide a frame to guide toward the most relevant molecular markers that would be affected during tumor development and could be biomarkers of PDAC cachexia. Inhibiting oxidative stress to reverse muscle wasting is a strategy currently under development (*Penna et al., 2020*; *Zhang et al., 2023*), but it is important to also consider the other side of oxidative stress, which drives the therapeutic response to chemo- and radiotherapy (*Abdel Hadi et al., 2021*; *Zhang et al., 2016*). In addition, our study points to the mitochondrial lipid metabolism as a promising target in PDAC cachexia, which was already suggested in other cancers with FAO inhibition by etomoxir (*Fukawa et al., 2016*) or trimetazidine (*Gatta et al., 2017*). Further investigation in this line is required, following our own data using the FAO inhibitor perhexiline which is able to cure chemotherapy-resistant tumors (*Reyes-Castellanos et al., 2023*) and has potential added benefit to limit chemotherapeutics toxicity (*Dhakal et al., 2023*). Another promising approach against muscle wasting in the clinic is physical activity which has multiple beneficial effects: maintenance of tumor mass, anti-inflammatory, antioxidant, and increase in mitochondrial quantity and quality, in line with the need for multimodal management of cachectic cancer patients (*Barbieri et al., 2015*; *Beltrà et al., 2021*; *Penna et al., 2020*; *Rygiel et al., 2016*; *Stubbins et al., 2020*).

**Table 1.** Comparison of mitoXplorer heatmaps for transcriptomics and proteomics analyses (related to *Figure 6—figure supplement 4*, *Figure 7—figure supplement 4*).

| Regulation in KIC muscle versus control | Genes/proteins | Pathways |
|---|---|---|
| Upregulated both in transcriptomics and proteomics analyses | Sirt2 | Mitochondrial signaling |
| Downregulated both in transcriptomics and proteomics analyses | Eno3<br>Pgam2<br>Mrpl19<br>Dlst<br>Pdp1<br>Dnaja1<br>Cyb5b | Glycolysis<br>Glycolysis<br>Translation<br>Tricarboxylic acid cycle<br>Pyruvate metabolism<br>Protein stability and degradation<br>Metabolism of vitamins and cofactors |
| Upregulated in transcriptomics and downregulated in proteomics analyses | Pcca<br>Gpx4<br>Gstp1<br>Aldh7a1<br>Aldh1l1 | Amino acid metabolism<br>ROS defense<br>ROS defense<br>Pyruvate metabolism<br>Folate and pterin metabolism |
| Downregulated in transcriptomics and upregulated in proteomics analyses | Mrpl45<br>Coq3<br>Nme1 | Translation<br>Ubiquinone biosynthesis<br>Nucleotide metabolism |

## Innovation

Mitochondrial dysfunction is a well-established feature of cachexia but was not explored in PDAC-induced sarcopenia. We characterized the KIC model through detailed functional, morphological, and omics-based assessment of muscle, demonstrating that tumor-driven muscle atrophy is associated with strong mitochondrial metabolic defects that are not limited to carbohydrate, protein, and redox metabolism, but also concern lipid and nucleic acid metabolism. Thus, this work provides a framework to guide toward the most relevant molecular markers that could be targeted in the clinic to limit PDAC-induced cachexia.

# Materials and methods

Electronic laboratory notebook was not used.

## Genetic mouse model of pancreatic cancer

We used male mice developing spontaneously a PDAC tumor, the *LSL-Kras$^{G12D}$;Ink4a/Arf$^{fl/fl}$;Pdx1-Cre* (KIC) mice, and *LSL-Kras$^{G12D}$;Ink4a/Arf$^{fl/fl}$* (KI) as control littermate mice (*Aguirre et al., 2003*). Body weight was measured weekly from the age of 4 weeks to the sacrifice at 9–11 weeks of age. Pancreas, liver, spleen, gonadal adipose tissue, and both gastrocnemius muscles were harvested and weighted. A piece of each organ and one gastrocnemius muscle were fast frozen in isopentane cooled on dry ice, and stored at −80°C for RNA and protein analysis. Considering that oxidative and glycolytic fibers are unevenly distributed in the gastrocnemius muscle, we paid particular attention to always using the same portion of the muscle for further molecular analyses. The second gastrocnemius muscle was used to extract mitochondria ex vivo for oxygraphy experiments (whole to avoid biases related to uneven distribution of oxidative and glycolytic fibers as explained before), or fast frozen and stored at −80°C for histological analysis. A piece of pancreas was fixed in 4% paraformaldehyde (PFA) and embedded in paraffin for histological analyses. All animal care and experimental procedures were performed with the approval of the Animal Ethics Committee of Aix-Marseille University under reference #21847-2019083116524081 for mice housed at the CRCM PSEA animal facility or reference #20423-2019042913133817v2 for mice housed at the CRMBM animal facility.

## Noninvasive MR investigation of gastrocnemius muscle function

Gastrocnemius muscle function was assessed noninvasively and longitudinally in five KIC mice before (7–8 weeks of age) and after (9–11 weeks) tumor development, and in seven age-matched control mice. The gastrocnemius muscle was chosen because it is easily accessible for MR antenna and preferentially activated using our methods (*Giannesini et al., 2010*). Investigations were performed with a homebuilt cradle designed to be operational inside the 7-Tesla horizontal magnet of a preclinical 70/16 PharmaScan MR scanner (Bruker, Karlsruhe, Germany). This cradle is similar to that described previously (*Giannesini et al., 2010*). Briefly, it allows (i) to elicit contraction of the left gastrocnemius muscle by transcutaneous electrostimulation, (ii) to measure the resulting contractile force with a dedicated ergometer composed of a foot pedal coupled to a force transducer, (iii) to perform multi-modal MR measurements, and (iv) to maintain prolonged anesthesia by gas inhalation (1.5–2% isoflurane mixed in air) with monitoring of breath rate and regulation of the animal's body temperature at physiological level throughout a feedback loop. Fourteen consecutive non-contiguous axial slices (1 mm thickness; 0.25 mm spaced) covering the whole left hindlimb were selected. Rapid acquisition relaxation-enhanced images of these slices (8 echoes; 5000 ms repetition time; 35 ms TE; 20×20 mm$^2$ field of view; 0.078×0.078 mm$^2$ spatial resolution) were acquired at rest. Muscle function was evaluated throughout a tetanic electrostimulation train (150 Hz; 0.5 s duration) and during a bout of exercise consisting in 6 min of repeated isometric contractions electrically induced at a frequency of 1 Hz. Contractile force was recorded and processed using the Powerlab 35 series system driven by the LabChart v8.1 software (AD Instruments, Oxford, UK). The digital signal was converted to force according to a linear calibration curve and expressed in N. $^{31}$P-MR spectra (15 accumulation; 2000 ms repetition time) from the gastrocnemius muscle region were continuously acquired at rest (4 min), during the 6 min exercise bout, and during the following 16 min recovery period. MR data were processed using custom software code written in Python. For each MR image, region of interest was manually outlined so that the corresponding cross-sectional area of the gastrocnemius muscle was

measured. Level of PCr and pH were obtained from $^{31}$P-MR spectra by a time-domain fitting routine using the AMARES algorithm (*Vanhamme, 1997*). The time constant of $\tau$ PCr was determined by fitting the PCr time changes during the post-exercise recovery period to a single exponential curve with a least mean-squared algorithm (*Kemp et al., 2015*): $\tau$ PCr = $-t/\ln(PCr_t/\Delta PCr)$, where $\Delta$PCr is the extent of PCr depletion measured at the start of the recovery period. The maximal oxidative capacity was calculated as the ratio between the basal PCr level and $\tau$ PCr (*Prompers et al., 2014*).

## Antibodies and reagents

Total OXPHOS rodent antibodies cocktail was from Abcam (mouse ab110413, RRID:AB_2629281). Mito-fusin1+2, laminin-2α, and MuRF1 antibodies were from Abcam (mouse ab57602 RRID:AB_2142624, rat ab11576 RRID:AB_298180, and mouse ab201941, respectively). β-Actin antibody was from Sigma-Aldrich (mouse A5316 RRID:AB_476743). Antibodies against myosin heavy chain (MHC)-I (mouse M8421), MHC-IIa (mouse SC-71-C), and MHC-IIb (mouse BF-F3-C) were obtained from the Developmental Studies Hybridoma Bank (DSHB), created by the NICHD of the NIH and maintained at The University of Iowa, Department of Biology, Iowa City, IA, USA. The secondary antibody used for western blotting was from Southern Biotech (Goat anti-mouse IgG conjugated to horseradish-peroxidase (HRP), 1030-05 RRID:AB_2619742). The blocking Goat anti-mouse IgG(H+L) was from Jackson Laboratory (115-007-003 RRID:AB_2338476). Secondary antibodies used for immunofluo-rescent staining, Goat anti-mouse Alexa Fluor 350 IgG2b (A-21140 RRID:AB_2535777), Goat anti-mouse Alexa Fluor 488 IgG1 (A-21121 RRID:AB_2535764), Goat anti-mouse Alexa Fluor 488 (A-11029 RRID:AB_2534088), Goat anti-mouse Alexa Fluor 594 IgM (A-21044 RRID:AB_2535713), Goat anti-rat Alexa Fluor 594 (A-11007 RRID:AB_10561522) were all from Invitrogen. DAPI reagent was from Thermo Fisher Scientific (62248).

## Hematoxylin, phloxine, and saffron staining

Gastrocnemius muscle cryosections (see below) or microtome sections of deparaffinized and rehy-drated pancreas (5-µm-thick) were stained according to a standard protocol. Briefly, the slides were incubated in Mayer's hematoxylin (Sigma-Aldrich 109249) during 3 min, fast washed in Ethanol 96%-HCl 1% then in water, then submitted to the bluing step during 3 min in sodium bicarbonate buffer (0.1% in water) before to be stained during 30 s with 2.5% phloxine solution (Sigma-Aldrich P2759), washed and stained 30 s in saffron in alcoholic solution (RAL Diagnostics 361500-1000) before final dehydration and mounting. Pictures of the whole slides were acquired with the Axio Imager 2 (ZEISS).

## Immunofluorescence microscopy

Cryosections of frozen gastrocnemius muscle (10 µm) were done at –22°C on a cryostat, collected on Superfrost plus glass slides, dried at 37°C for 10 min, fixed in 4% PFA for 10 min before being permeabilized in phosphate-buffered saline (PBS)1X+Triton 0.2% (PBS-T) for 30 min. Sections were blocked and saturated with PBS-T+2% BSA+2% Goat serum for 1 hr, washed three times in PBS1X for 5 min, blocked with Goat anti-mouse IgG(H+L) (1:1000) for 1 hr if necessary, washed three times in PBS1X for 5 min before to be incubated with primary antibody for 2 hr at room temperature (RT). After three washes in PBS1X for 5 min, sections were incubated with adapted secondary antibodies for 1 hr at RT, washed in PBS, and incubated with DAPI for nuclear staining before final wash and mounting with Prolong Gold Antifade mounting medium from Thermo Fisher Scientific (P36934). Pictures of the whole slides were acquired with the Axio Imager 2 (ZEISS), and the cross-sectional area, fiber Feret's diameter (longest distance [µm] between any two points within a given fiber), and MuRF1-positive-stained area in muscle sections were automatically measured by ImageJ software. The following primary antibody dilutions were used: total OXPHOS rodent antibodies cocktail (1:200), anti-Laminin 2α (1:200), anti-MuRF1 (1:200), anti-MHC-I (1:500), anti-MHC-IIa (1:600), anti-MHC-IIb (1:1000). Adapted Alexa Fluor-conjugated secondary antibodies were used at 1:500 dilutions.

## Transmission electron microscopy

Mice were transcardially perfused with cold PBS (10 mM, pH 7.4), followed by 2% glutaraldehyde solution in 0.1 M cacodylate buffer (pH 7.4), under ketamine (150 mg/kg) and xylazine (20 mg/kg) anes-thesia. After necropsia, gastrocnemius muscle samples were immediately fixed in a 2% glutaraldehyde

solution in 0.1 M cacodylate buffer (pH 7.4), postfixed in 1% osmium tetroxide in 0.1 M cacodylate buffer, dehydrated in increasing concentration of ethanol followed by acetone, and embedded in Epon. Sections (1-μm-thick) were stained with toluidine blue to verify the orientation of the muscle tissue prior to the ultrathin sectioning. Ultrathin sections were cut in longitudinal or transverse orientation using an Ultracut UC7 ultramicrotome (Leica) and mounted on copper carbon-formvar-coated grids. Uranyl acetate and lead citrate-stained sections were imaged using an FEI G2 electron microscope (FEI) with 200 KeV. Digital micrographs were captured using a Veleta (Olympus) digital camera. Individual mitochondria from CTRL (n=3) and KIC (n=3) mice were manually traced in longitudinal and transverse orientations using ImageJ (NIH), to quantify the following morphological and shape descriptors: area (in $\mu m^2$), circularity ($4\pi \cdot$[surface area/perimeter$^2$]), and Feret's diameter (longest distance [μm] between any two points within a given mitochondrion).

## Protein isolation and western blot analysis

Gastrocnemius muscle samples were lysed in lysis buffer (10% N-deoxycholate, 0.1% SDS, 1% Triton X-100, 10 mM Tris pH 8, 140 mM NaCl, phosphatases/proteases inhibitors [Sigma-Aldrich], 1% phenyl-methylsulfonyl fluoride, 1 mM sodium fluoride, 100 μM sodium orthovadanate, 40 mM beta glycerophosphate) in beads tubes from Precellys lysing kit for hard tissue (P000917-LYSK0-A; 3× Cycle 1500 rpm, 15 s of mix, 10 s of rest) and incubated at 4°C for 30 min to solubilize proteins before centrifugation (10,000×g, 10 min, 4°C). Supernatants were collected and protein concentration evaluated using Bio-Rad Protein assay. Proteins were diluted in 1× Laemmli sample buffer and heated at 95°C for 10 min in the presence of β-mercaptoethanol (Sigma) (except for OXPHOS cocktail antibody according to the manufacturer's instructions). Whole protein extracts (25–75 μg per lane) were resolved by SDS-polyacrylamide gel electrophoresis using a 10% acrylamide gel, and transferred onto 0.2 μm nitrocellulose membranes (GE Healthcare). To check for protein transfer, the membranes were stained with Ponceau Red (in most of cases) or with amidoblack. After a blocking step (5% non-fat milk in TBS overnight [O/N] at 4°C), membranes were incubated with primary antibodies diluted in 5% non-fat milk in TBS 0.1% Tween O/N at 4°C. Following dilutions of primary antibodies were used: anti-OXPHOS cocktail (1:500), anti-MuRF1 (1:200), anti-β-actin (1:10,000). After three washes in TBS 0.1% Tween, membranes were incubated 2 hr at RT with HRP-conjugated secondary antibodies at 1:5000 in 5% non-fat milk in TBS 0.1% Tween. ECL protein detection (Millipore) was performed with a Fusion Fx7 chemiluminescent imager, and quantification of proteins of interest was done by densitometry using ImageJ software and normalized to Ponceau Red densitometry.

## Mitochondrial isolation

Mitochondria were extracted from fresh entire gastrocnemius muscles. Briefly, muscles were collected, cut in Petri dishes, and incubated in 5 mL of Isolation Mitochondrial Buffer 1 (IBM1; BSA 0.2%, EDTA 10 mM, KCl 50 mM, Tris/HCl 50 mM, sucrose 67 mM, pH 7.4) with 0.05% trypsin for 1 hr under agitation at 4°C. After a brief centrifugation (3 min, 200×g, 4°C), supernatant was removed and muscle pieces collected and homogenized in IBM1 using Teflon glass pestle homogenizer in ice. Mitochondria were isolated via differential centrifugation. Briefly, the homogenate was centrifuged (10 min, 1200×g, 4°C) and the supernatant decanted into a new tube. This fraction was centrifuged again (10 min, 8000×g, 4°C) to obtain a mitochondrial pellet. Supernatant which contains soluble proteins was carefully removed, and the mitochondrial pellet was washed and resuspended carefully in 200 μL of Isolation Mitochondrial Buffer 2 (IBM2; D-Mannitol 200 mM, EGTA 5 mM, Tris/HCl 10 mM, sucrose 70 mM, pH 7.4) then diluted in 6 mL of IBM2 followed by a centrifugation to purify mitochondria. This step was repeated a second time before resuspension of mitochondrial pellet in 50–100 μL of IBM2 at 4°C. BCA assay was performed with a spectrophotometer to evaluate mitochondrial protein concentration before oxygraphic measurements.

## Oxygraphic respiration assays on isolated mitochondria

Seahorse XF24 instrument (Agilent Technologies) was equilibrated at 37°C and XF Sensor cartridge hydrated O/N. Mitochondria isolated from gastrocnemius muscle (2.5 μg equivalent proteins) were diluted in 50 μL final volume of 1× Mitochondrial Assay Solution (MAS; D-Mannitol 200 mM, EGTA 5 mM, Tris/HCl 10 mM, sucrose 70 mM, pH 7.4), and plated in Seahorse plate wells after supplementation with ADP (5 mM), succinate (10 mM), and rotenone (2 μM) for the coupling assay, or pyruvate

(5 mM), malate (1 mM), and FCCP (4 µM) for the electron flow assay (*Boutagy et al., 2015*). The plate was centrifuged (20 min, 2000×*g*, RT), 450 µL of 1× MAS containing suitable substrates was added to each well, and the plate was incubated in a non-CO₂ incubator at 37°C for 8–10 min. The XF cartridge was loaded during plate centrifugation, with reagents diluted in 1× MAS containing suitable substrates: for the coupling assay, oligomycin, FCCP, and antimycin A (4 µM, 4 µM, and 8 µM final concentration in the well, respectively); for the electron flow assay, rotenone, succinate, antimycin A, and ascorbate/TMPD (2 µM, 5 mM, 8 µM, 10 mM/100 µM final concentration, respectively). After cartridge calibration, the plate containing mitochondria was introduced into the machine and the assay continued using a protocol of 1 min mixing, 20 s waiting, 2 min measurement cycles done two times after calibration and after each injection.

## Total DNA isolation from gastrocnemius skeletal muscle

Total DNA (genomic and mtDNA) was isolated from ≈20 to 30 mg of gastrocnemius skeletal muscle. Briefly, tissues were lysed O/N at 50°C under agitation in DNA lysis buffer (0.2% SDS, 100 mM Tris pH 8, 200 mM NaCl, 500 nM EDTA) with 200 µg/mL Proteinase K RNA grade (Invitrogen, 25 530-049) followed by RNAse incubation (Thermo Fisher, AM2294) to remove RNA contamination. Samples were centrifuged (12,000×*g*, 10 min, RT) and the supernatant was transferred and mixed in equal volume of absolute ethanol to allow DNA precipitation. Samples were centrifuged (12,000×*g*, 10 min, RT), supernatant was removed and DNA pellet resuspended in RNAse/DNAse-free water. DNA concentration and purity were assessed using spectrophotometer.

## mtDNA copy number analysis

mtDNA copy number was quantified by qPCR on total DNA from gastrocnemius muscle using MxPro Mx3000P real-time PCR Analyzer (Agilent Technologies/Stratagene) and determining the mtDNA/nDNA ratio. Primers for mitochondrial genes were: *mt12S* (mt12S-F ACCGCGGTCATACGATTAAC, mt12S-R CCCAGTTTGGGTCTTAGCTG) and *mtCOX3* (mtCOX3-F GGTCTCGGAAGTATTTTTCTTTGC, mtCOX3-R CAGCAGCCTCCTAGATCATGTG). *NDUFV1* gene was used as nuclear gene for normalization (gNDUFV1-F CTTCCCCACTGGCCTCAAG, gNDUFV1-R CCAAAACCCAGTGATCCAGC).

## Transcriptomic analysis by bulk RNA sequencing

### Total RNA isolation from gastrocnemius skeletal muscle

Total RNA was isolated from ≈25 mg of the gastrocnemius skeletal muscle using both TRIzol (Invitrogen) and the QIAGEN total RNA isolation kit (QIAGEN, ref 74004) to avoid protein and extracellular matrix accumulation in the columns. Briefly, muscle piece was lysed in 600 µL RLT Buffer (from QIAGEN kit) with 1% β-mercaptoethanol using beads tube from Precellys lysing kit for hard tissue (P000917-LYSK0-A; 3× Cycle 1500 rpm, 15 s of mix, 10 s of rest) and directly centrifuged 3 min at 10,000×*g* at RT. Supernatant was transferred in 400 µL of TRIzol and incubated for 5 min at RT followed by addition of 150 µL of chloroforme. Tubes were carefully mixed and incubated 3 min at RT before centrifugation (12,000×*g*, 5 min, 4°C). The transparent upper phase containing RNAs was transferred in a tube containing 500 µL of 70% ethanol, mixed and transferred inside QIAGEN RNA isolation kit columns according to the manufacturer's instructions followed until the end of extraction. Samples were treated with RNase-free DNase on QIAGEN spin-columns to remove DNA contamination. RNA integrity and concentration were assessed using the Agilent 2100 Bioanalyzer (Agilent Technologies, Palo Alto, CA, USA). The average RIN (RNA integrity number) values for all samples were comprised between 9.3 and 10, ensuring a high quality of isolated RNAs.

### Bulk RNA sequencing

The preparation of mRNA libraries was realized following the manufacturer's recommendations (kapa mRNA HyperPrep from ROCHE). Final samples pooled library prep were sequenced on ILLUMINA NovaSeq 6000 with S1-200 cycles cartridge (2×1600 millions of 100 bases reads), corresponding to 2×30 millions of reads per sample after demultiplexing.

### Bulk RNA sequencing data analysis

Quality control was performed on the fastq files using FastQC (v0.11.9) (http://www.bioinformatics.babraham.ac.uk/projects/fastqc). To map the sequenced reads to the mouse reference genome

(GRCm39 from NCBI), we made use of STAR (v2.7.3a) (*Al Tanoury et al., 2021*). From these mapped reads, gene counts were then quantified using featureCounts (v2.0.1) (*Al Tanoury et al., 2021*). Starting from the raw gene counts, normalization and differential expression analysis was then performed using DESeq2 (v1.22.2) (*Al Tanoury et al., 2021*). Resulting from the differential expression analysis, genes were defined as significantly deregulated when the false discovery rate (FDR), related to the fold change, was lower than 0.05.

## Proteomic analysis by mass spectrometry

### Sample preparation and mass spectrometry

Proteomes from gastrocnemius muscles prepared from control and KIC mice were compared by label-free quantitative mass spectrometry analysis. Briefly, 15 µg of each lysate prepared from muscle was loaded on NuPAGE 4–12% Bis-Tris acrylamide gels (Life Technologies) and run at 80 V for 8 min to stack proteins in a single band. Bands were cut from the gels following Imperial Blue staining (Thermo Fisher Scientific) and were submitted to a classical in-gel trypsin digestion after protein reduction and cysteine alkylation with iodoacetamide (*Shevchenko et al., 1996*). Peptides were extracted from the gel and dried under vacuum. Each sample was reconstituted with 0.1% trifluoroacetic acid in 2% acetonitrile and analyzed thrice by liquid chromatography (LC)-tandem mass spectrometry (MS/MS) using an Orbitrap Fusion Lumos Tribrid Mass Spectrometer (Thermo Fisher Scientific, San Jose, CA, USA) online with a nanoRSLC Ultimate 3000 chromatography system (Thermo Fisher Scientific, Sunnyvale, CA, USA). For each run, peptides were first concentrated on a pre-column (C18 PepMap100, 2 cm × 100 µm I.D., 100 Å pore size, 5 µm particle size) and washed in water containing 0.1% trifluoroacetic acid. In a second step, peptides were separated on a reverse phase LC EASY-Spray C18 column (PepMap RSLC C18, 50 cm × 75 µm I.D., 100 Å pore size, 2 µm particle size) at 300 nL/min flow rate with the following parameters: after column equilibration using 3% of solvent B (20% water - 80% acetonitrile - 0.1% formic acid), peptides were eluted from the analytical column by a two-step linear gradient (2.4–22% acetonitrile/$H_2O$; 0.1% formic acid for 130 min and 22–32% acetonitrile/$H_2O$; 0.1% formic acid for 15 min). MS analysis was performed following peptide ionization in the EASY-Spray Nano-source in front of the Orbitrap Fusion Lumos Tribrid under a spray voltage set at 2.2 kV and the capillary temperature at 275°C. The Orbitrap Lumos was used in data-dependent mode to switch consistently between MS and MS/MS. Time between Masters Scans was set to 3 s. MS spectra were acquired with the Orbitrap in the range of m/z 400–1600 at an FWHM resolution of 120,000 measured at 400 m/z. AGC target was set at 4.0e5 with a 50 ms maximum injection time. For internal mass calibration the 445.120025 ion was used as lock mass. The more abundant precursor ions were selected and collision-induced dissociation fragmentation was performed in the ion trap to have maximum sensitivity and yield a maximum amount of MS/MS data. Number of precursor ions was automatically defined along run in 3 s windows using the 'Inject Ions for All Available parallelizable time option' with a maximum injection time of 300 ms. The signal threshold for an MS/MS event was set to 5000 counts. Charge state screening was enabled to exclude precursors with 0 and 1 charge states. Dynamic exclusion was enabled with a repeat count of 1 and duration of 60 s.

### Data processing protocol

Relative intensity-based label-free quantification (LFQ) was processed using the MaxLFQ algorithm from the freely available MaxQuant computational proteomics platform, version 1.6.3.4 (*Cox et al., 2014*; *Cox and Mann, 2008*). The acquired raw LC Orbitrap MS data were first processed using the integrated Andromeda search engine (*Cox et al., 2011*). Spectra were searched against the *Mus musculus* proteome extracted from UniProt on December 2020 and containing 55508 entries (UP000000589). The FDR at the peptide and protein levels were set to 1% and determined by searching a reverse database. For protein grouping, all proteins that cannot be distinguished based on their identified peptides were assembled into a single entry according to the MaxQuant rules. The statistical analysis was done with Perseus program (v1.6.14.0) from the MaxQuant environment (https://www.maxquant.org/). Quantifiable proteins were defined as those detected in above 70% of samples in one condition or more. Protein LFQ normalized intensities were base 2 logarithmized to obtain a normal distribution. Missing values were replaced using data imputation by randomly selecting from a normal distribution centered on the lower edge of the intensity values that simulates signals of low abundant proteins using default parameters (a downshift of 1.8 standard deviation

and a width of 0.3 of the original distribution). To determine whether a given detected protein was specifically differential, a two-sample t-test was done using permutation-based FDR-controlled at 0.05 and employing 250 permutations. The p-value was adjusted using a scaling factor s0 with a value of 1 (*Tusher et al., 2001*). From the results of the statistical analysis proteins with FDR, related to the fold change, lower than 0.05 were then defined as dysregulated. This analysis included four biological replicates of control mice and five KIC mice. Each biological replicate was injected twice on the instruments.

## Enrichment analysis

Enrichment analysis was performed both on the differentially expressed genes and proteins through the use of IPA (QIAGEN Inc). Differentially expressed genes and proteins data were also uploaded on the mitoXplorer web platform to explore mitochondrial dynamics in the different conditions (*Yim et al., 2020*). Heatmaps were generated on mitoXplorer web platform.

## Statistical analysis

We used a Mann-Whitney non-parametric test to compare two experimental groups. One-way or two-way ANOVA, followed by the appropriated multiple comparison test, was used when comparing more than two experimental groups. In all figures, data are presented as mean ± standard error of the mean (SEM) and exact n values used to calculate the statistics are indicated. All statistical analyses were performed using GraphPad Prism 8 (GraphPad Software). p-values<0.05 were considered statistically significant.

## Acknowledgements

We thank Marc Bartoli for recommendations in murine muscle atrophy exploration, Jérôme Robin and Arnaud Mourier for advices in isolation of active mitochondria, Sofia Vikstrom and May Sanderhoff (Seahorse/Agilent Support Scientists) for help in Seahorse experiments on isolated mitochondria, Karim Sari and Régis Vitestelle for assistance with the use of the PSEA animal housing facility, and Nuno Luis, Jérôme Avellaneda, Frank Schnorrer, and Thomas Rival for their suggestions and comments on the results. We are grateful to members of the PiCSL-FBI core facility (IBDM, AMU-Marseille, France) belonging to the France-BioImaging national research infrastructure for the electron microscopy experiments. This work benefited from equipment and services from the iGenSeq core facility (Genotyping and sequencing), at Institut du Cerveau et de la Moelle épinière (ICM, Paris, France). Proteomic analyses were performed at the mass spectrometry facility of Marseille Proteomics (https://marseille-proteomique.univ-amu.fr/) supported by IBISA (Infrastructures Biologie Santé et Agronomie), Plateforme Technologique Aix-Marseille, the Cancéropôle PACA, the Provence-Alpes-Côte d'Azur Région, the Institut Paoli-Calmettes and the Centre de Recherche en Cancérologie de Marseille, Fonds Européen de Developpement Régional, and Plan Cancer.

This work was supported by Institut National de la Santé et de la Recherche Médicale, Centre National de la Recherche Scientifique, French Agence Nationale de la Recherche (ANR-18-CE45-0016-03 'MITODYNAMICS'). This work also benefited from grants from Cancéropôle Provence Alpes Côte d'Azur (PACA) and Fédération GEFLUC for the acquisition of the Seahorse XFe24 device. CRMBM is a laboratory member of France Life Imaging (FLI) network (grant ANR-11-INBS-0006). TG and FM were supported by the ANR, and GR-C by the CONACYT (Mexico) and Ligue Nationale Contre le Cancer.

## Additional information

### Funding

| Funder | Grant reference number | Author |
|---|---|---|
| Agence Nationale de la Recherche | ANR-18-CE45-0016-03 | Bianca H Habermann<br>Benoit Giannesini<br>Alice Carrier |

| Funder | Grant reference number | Author |
|--------|------------------------|--------|

The funders had no role in study design, data collection and interpretation, or the decision to submit the work for publication.

## Author contributions

Tristan Gicquel, Resources, Data curation, Formal analysis, Validation, Investigation, Visualization, Methodology, Writing - original draft, Project administration, Writing - review and editing; Fabio Marchiano, Formal analysis, Validation; Gabriela Reyes-Castellanos, Conceptualization, Funding acquisition; Stephane Audebert, Luc Camoin, Formal analysis, Validation, Investigation, Visualization, Methodology; Bianca H Habermann, Conceptualization, Formal analysis, Funding acquisition, Validation, Project administration; Benoit Giannesini, Alice Carrier, Conceptualization, Formal analysis, Supervision, Funding acquisition, Validation, Investigation, Visualization, Methodology, Writing - original draft, Project administration, Writing - review and editing

## Author ORCIDs

Luc Camoin ⓘ https://orcid.org/0000-0002-1230-4787
Bianca H Habermann ⓘ https://orcid.org/0000-0002-2457-7504
Alice Carrier ⓘ https://orcid.org/0000-0001-6599-9642

## Ethics

All animal care and experimental procedures were performed with the approval of the Animal Ethics Committee of Aix-Marseille University under reference #21847-2019083116524081 for mice housed at the CRCM PSEA animal facility or reference #20423-2019042913133817v2 for mice housed at the CRMBM animal facility.

Reviewer #1 (Public Review): https://doi.org/10.7554/eLife.93312.2.sa1
Reviewer #2 (Public Review): https://doi.org/10.7554/eLife.93312.2.sa2
Author response https://doi.org/10.7554/eLife.93312.2.sa3

# Additional files

## Supplementary files

• MDAR checklist

## Data availability

Gene expression data were deposited in the Gene Expression Omnibus repository at NCBI (accession number GSE226898). The mass spectrometry proteomics data, including search results, have been deposited to the ProteomeXchange Consortium (https://www.proteomexchange.org/) (*Deutsch et al., 2020*) via the PRIDE (*Perez-Riverol et al., 2019*) partner repository with the dataset identifier PXD036752. Datasets were deposited into Mendeley data repository (https://doi.org/10.17632/w25cc5pv34.1).

The following datasets were generated:

| Author(s) | Year | Dataset title | Dataset URL | Database and Identifier |
|-----------|------|---------------|-------------|-------------------------|
| Marchiano F, Gicquel T, Carrier A, Habermann B | 2024 | Integrative study of mitochondrial dysfunction in murine skeletal muscle during pancreatic cancer cachexia | https://www.ncbi.nlm.nih.gov/geo/query/acc.cgi?acc=GSE226898 | NCBI Gene Expression Omnibus, GSE226898 |
| Carrier A, Audebert S | 2024 | Integrative study of mitochondrial dysfunction in skeletal muscle during pancreatic cancer cachexia | https://proteomecentral.proteomexchange.org/cgi/GetDataset?ID=PXD036752 | ProteomeXchange, PXD036752 |

*Continued on next page*

Continued

| Author(s) | Year | Dataset title | Dataset URL | Database and Identifier |
|---|---|---|---|---|
| Giannesini B, Carrier A | 2024 | Integrative study of skeletal muscle mitochondrial dysfunction in a murine pancreatic cancer-induced cachexia model | https://data.mendeley.com/datasets/w25cc5pv34/1 | Mendeley Data, 10.17632/w25cc5pv34.1 |

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
