## [Editor Report · eLife assessment]

This **useful** study uses a mouse model of pancreatic cancer to examine mitochondrial mass and structure in atrophying muscle along with aspects of mitochondrial metabolism in the same tissue. Most relevant are the **solid** transcriptomics and proteomics approaches to map out related changes in gene expression networks in muscle during cancer cachexia.

---

## [Referee Report · Reviewer #1 (Public Review)]

Summary:

This important study provides a comprehensive evaluation of skeletal muscle mitochondrial function and remodeling in a genetically engineered mouse model of pancreatic cancer cachexia. The study builds upon and extends previous findings that implicate mitochondrial defects in the pathophysiology of cancer cachexia. The authors demonstrate that while the total quantity of mitochondria from skeletal muscles of mice with pancreatic cancer cachexia is similar to controls, mitochondria were elongated with disorganized cristae, and had reduced oxidative capacity. The mitochondrial dysfunction was not associated with exercise-induced metabolic stress (insufficient ATP production), suggesting compensation by glycolysis or other metabolic pathways. However, mitochondrial dysfunction can lead to increased production of ROS/oxidative stress and would be expected to interfere with carbohydrate and lipid metabolism, events that are linked to cancer-induced muscle loss. The data are convincing and were collected and analyzed using state-of-the-art techniques, with unbiased proteomics and transcriptomics analyses supporting most of their conclusions.

Additional Strengths:

The authors utilize a genetically engineered mouse model of pancreatic cancer which recapitulates key aspects of human PDAC including the development of cachexia, making the model highly appropriate and translational.

The authors perform transcriptomic and proteomics analyses on the same tissue, providing a comprehensive analysis of the transcriptional networks and protein networks changed in the context of PDAC cachexia.

Weaknesses:

The authors refer to skeletal muscle wasting induced by PDAC as sarcopenia. However, the term sarcopenia is typically reserved for the loss of skeletal muscle mass associated with aging.

In Figure 2, the MuRF1 IHC staining appears localized to the extracellular space surrounding blood vessels and myofibers-which causes concern as to the specificity of the antibody staining. MuRF1, as a muscle-specific E3 ubiquitin ligase that degrades myofibrillar proteins, would be expected to be expressed in the cytosol of muscle fibers.

Disruptions to skeletal muscle metabolism in PDAC mice are predicted based on mitochondrial dysfunction and the transcriptomic and proteomics data. The manuscript could therefore be strengthened by additional measures looking at skeletal muscle metabolites, or linking the findings to previous work that has looked at the skeletal muscle metabolome in related models of PDAC cachexia (Neyroud et al., 2023).

---

## [Referee Report · Reviewer #2 (Public Review)]

The present work analyzed the mitochondrial function and bioenergetics in the context of cancer cachexia induced by pancreatic cancer (PDAC). The authors used the KIC transgenic mice that spontaneously develop PDAC within 9-11 weeks of age. They deeply characterize bioenergetics in living mice by magnetic resonance (MR) and mitochondrial function/morphology mainly by oxygraphy and imaging on ex vivo muscles. By MR they found that phosphocreatine resynthesis and maximal oxidative capacity were reduced in the gastrocnemius muscle of tumor-bearing mice during the recovery phase after 6 minutes of 1 Hz electrical stimulation while pH was reduced in muscle during the stimulation time. By oxygraphy, the authors showed a decrease in basal respiration, proton leak, and maximal respiration in tumor-bearing mice that was associated with the decrease of complex I, II, and IV activity, a reduction of OXPHOS proteins, mitochondrial mass, mtDNA, and to several morphological alterations of mitochondrial shape. The authors performed transcriptomic and proteomic analyses to get insights into mitochondrial defects in the muscles of PDAC mice. By IPA analyses on transcriptomics, they found an increase in the signature of protein degradation, atrophy, and glycolysis and a downregulation of muscle function. Focusing on mitochondria they showed a downregulation mainly in OXPHOS, TCA cycle, and mitochondrial dynamics genes and upregulation of glycolysis, ROS defense, mitophagy, and amino acid metabolism. IPA analysis on proteomics revealed major changes in muscle contraction and metabolic pathways related to lipids, protein, nucleotide, and DNA metabolism. Focusing on mitochondria, the protein changes mainly were related to OXPHOS, TCA cycle, translation, and amino acid metabolism.

The major strength of the paper is the bioenergetics and mitochondrial characterization associated with the transcriptomic and proteomic analyses in PDAC mice that confirmed some published data of mitochondrial dysfunction but underlined some novel metabolic insights such as nucleotide metabolism.

There are minor weaknesses related to some analyses on mitochondrial proteins and to the fact that proteomic and transcriptomic comparison may be problematic in catabolic conditions because some gene expression is required to maintain or re-establish enzymes/proteins that are destroyed by the proteolytic systems (including the autophagy proteins and ubiquitin ligases). The authors should consider the following points.

Point1. The authors used the name sarcopenia as synonymous with muscle atrophy. However, sarcopenia clearly defines the disease state (disease code: ICD-10-CM (M62.84)) of excessive muscle loss and force drop during ageing (Ref: Anker SD et al. J Cachexia Sarcopenia Muscle 2016 Dec;7(5):512-514.). Therefore, the word sarcopenia must be used only when pathological age-related muscle loss is the subject of study. Sarcopenia can be present in cancer patients who also experience cachexia, however since the age of tumor-bearing mice in this study is 7-9 weeks old, the authors should refrain from using sarcopenia and instead replace it with the words muscle atrophy/ muscle wasting/muscle loss.

Point2. Most of the analyses of mitochondrial function are appropriate. However, the methodological approach to determining mitochondrial fusion and fission machinery shown in Fig. 5F is wrong. The correct way is to normalize the OPA1, MFn1/2 on mitochondrial proteins such as VDAC/porin. In fact, by loading the same amount of total protein (see actin in panel 5F) the difference between a normal and a muscle with enhanced protein breakdown is lost. In fact, we should expect a decrease in actin level in tumor-bearing mice with muscle atrophy while the blots clearly show the same level due to the normalization of protein content. Moreover, by loading the same amount of proteins in the gel, the atrophying muscle lysates become enriched in the proteins/organelles that are less affected by the proteolysis resulting in an artefactual increase. The correct way should be to lyse the whole muscle of control and tumor-bearing mice in an identical volume and to load in western blot the same volume between control cachectic muscles. Alternatively, the relative abundance of mitochondrial shaping proteins related to mitochondrial transmembrane or matrix proteins (mito mass) should compensate for the loading normalization. Because the authors showed elongated mitochondria despite mitophagy genes being up, fragmentation may be altered. Moreover, DNM1l gene is suppressed and therefore DRP1 protein must be analyzed. Finally, OPA 1 protein has different isoforms due to the action of proteases like OMA1, and YME1L that elicit different functions being the long one pro-fusion while the short ones do not. The authors must quantify the long and short isoforms of OPA1.

Point3. The comparison of proteomic and transcriptomic profiles to identify concordance or not is problematic when atrophy programs are induced. In fact, most of the transcriptional-dependent upregulation is to preserve/maintain/reestablish enzymes that are consumed during enhanced protein breakdown. For instance, the ubiquitin ligases when activated undergo autoubiquitination and proteasome degradation. The same happens for several autophagy-related genes belonging to the conjugation system (LC3, Gabarap), the cargo recognition pathways (e.g. Ubiquitin, p62/SQSTM1) and the selective autophagy system (e.g. BNIP3, PINK/PARKIN) and metabolic enzymes (e.g. GAPDH, lipin). Finally, in case identical amounts of proteins have been loaded in mass spec the issues rise in point 2 of selective enrichment should be considered. Therefore, when comparing proteomic and transcriptomic these issues should be considered in discussion.

---

## [Author Response]

eLife assessmentThis useful study uses a mouse model of pancreatic cancer to examine mitochondrial mass and structure in atrophying muscle along with aspects of mitochondrial metabolism in the same tissue. Most relevant are the solid transcriptomics and proteomics approaches to map out related changes in gene expression networks in muscle during cancer cachexia.

Response: We very much appreciate the positive feedback from the editors on our article and are delighted to have it published in eLife. Our sincere thanks to the Reviewers for their positive feedback on our work, and for their insightful and constructive comments.

**Reviewer #1 (Public Review):**
Summary:This important study provides a comprehensive evaluation of skeletal muscle mitochondrial function and remodeling in a genetically engineered mouse model of pancreatic cancer cachexia. The study builds upon and extends previous findings that implicate mitochondrial defects in the pathophysiology of cancer cachexia. The authors demonstrate that while the total quantity of mitochondria from skeletal muscles of mice with pancreatic cancer cachexia is similar to controls, mitochondria were elongated with disorganized cristae, and had reduced oxidative capacity. The mitochondrial dysfunction was not associated with exercise-induced metabolic stress (insufficient ATP production), suggesting compensation by glycolysis or other metabolic pathways. However, mitochondrial dysfunction can lead to increased production of ROS/oxidative stress and would be expected to interfere with carbohydrate and lipid metabolism, events that are linked to cancer-induced muscle loss. The data are convincing and were collected and analyzed using state-of-the-art techniques, with unbiased proteomics and transcriptomics analyses supporting most of their conclusions.Additional Strengths:The authors utilize a genetically engineered mouse model of pancreatic cancer which recapitulates key aspects of human PDAC including the development of cachexia, making the model highly appropriate and translational.The authors perform transcriptomic and proteomics analyses on the same tissue, providing a comprehensive analysis of the transcriptional networks and protein networks changed in the context of PDAC cachexia.Weaknesses:The authors refer to skeletal muscle wasting induced by PDAC as sarcopenia. However, the term sarcopenia is typically reserved for the loss of skeletal muscle mass associated with aging.

Response: We agree that the term sarcopenia initially refers to aged muscle, but its use has spread to other fields, including oncology (for example, in this article, which we quote: Mintziras I et al. Sarcopenia and sarcopenic obesity are significantly associated with poorer overall survival in patients with pancreatic cancer: Systematic review and meta-analysis. Int J Surg 2018;59:19-26). Actually, the term sarcopenia is now widely used in the literature and in the clinic to describe the loss of muscle mass and strength in cancer patients (see for example, this recent review: Papadopetraki A. et al. The Role of Exercise in Cancer-Related Sarcopenia and Sarcopenic Obesity. Cancers 2023;15;5856).

In Figure 2, the MuRF1 IHC staining appears localized to the extracellular space surrounding blood vessels and myofibers-which causes concern as to the specificity of the antibody staining. MuRF1, as a muscle-specific E3 ubiquitin ligase that degrades myofibrillar proteins, would be expected to be expressed in the cytosol of muscle fibers.

Response: We agree that MuRF1 IHC staining was also observed in the extracellular space, which was a surprise, for which we have no explanation to date.

Disruptions to skeletal muscle metabolism in PDAC mice are predicted based on mitochondrial dysfunction and the transcriptomic and proteomics data. The manuscript could therefore be strengthened by additional measures looking at skeletal muscle metabolites, or linking the findings to previous work that has looked at the skeletal muscle metabolome in related models of PDAC cachexia (Neyroud et al., 2023).

Response: We agree that our omics data could be strengthened by additional measures looking at skeletal muscle metabolites. It's an excellent suggestion to parallel the transcriptomic and proteomic data we obtained on the gastrocnemius muscle with the metabolomic data obtained by Neyroud et al. on the same muscle. These authors used another mouse model of PDAC than our KIC GEMM model, namely the allograft model implanting KPC cells (derived from the pancreatic tumor of KPC mice, another PDAC GEMM model) into syngeneic recipient mice. They carried out a proteomic study on the tibialis anterior muscle and a metabolomic study on the gastrocnemius muscle. Proteomics data identified in particular a KPC-induced reduction in the relative abundance of proteins annotating to oxidative phosphorylation, consistently with our data showing reduced mitochondrial activity pathways. Metabolomic data showed reduced abundance of many amino acids as expected, and of intermediates of the mitochondrial TCA cycle (malate and fumarate) in KPC-atrophied muscle consistently with reduced mitochondrial metabolic pathways that we illustrated. In contrast, metabolites that were increased in abundance included those related to oxidative stress and redox homeostasis, which is not surprising regarding the profound oxidative stress affecting atrophied muscle. Finally, we noted in Neyroud's metabolomic data the dysregulation of certain lipids and nucleotides in atrophied muscle, which is very interesting to relate to our study describing alterations in lipid and nucleotide metabolic pathways.

**Reviewer #2 (Public Review):**
The present work analyzed the mitochondrial function and bioenergetics in the context of cancer cachexia induced by pancreatic cancer (PDAC). The authors used the KIC transgenic mice that spontaneously develop PDAC within 9-11 weeks of age. They deeply characterize bioenergetics in living mice by magnetic resonance (MR) and mitochondrial function/morphology mainly by oxygraphy and imaging on ex vivo muscles. By MR they found that phosphocreatine resynthesis and maximal oxidative capacity were reduced in the gastrocnemius muscle of tumor-bearing mice during the recovery phase after 6 minutes of 1 Hz electrical stimulation while pH was reduced in muscle during the stimulation time. By oxygraphy, the authors showed a decrease in basal respiration, proton leak, and maximal respiration in tumor-bearing mice that was associated with the decrease of complex I, II, and IV activity, a reduction of OXPHOS proteins, mitochondrial mass, mtDNA, and to several morphological alterations of mitochondrial shape. The authors performed transcriptomic and proteomic analyses to get insights into mitochondrial defects in the muscles of PDAC mice. By IPA analyses on transcriptomics, they found an increase in the signature of protein degradation, atrophy, and glycolysis and a downregulation of muscle function. Focusing on mitochondria they showed a downregulation mainly in OXPHOS, TCA cycle, and mitochondrial dynamics genes and upregulation of glycolysis, ROS defense, mitophagy, and amino acid metabolism. IPA analysis on proteomics revealed major changes in muscle contraction and metabolic pathways related to lipids, protein, nucleotide, and DNA metabolism. Focusing on mitochondria, the protein changes mainly were related to OXPHOS, TCA cycle, translation, and amino acid metabolism.The major strength of the paper is the bioenergetics and mitochondrial characterization associated with the transcriptomic and proteomic analyses in PDAC mice that confirmed some published data of mitochondrial dysfunction but underlined some novel metabolic insights such as nucleotide metabolism.There are minor weaknesses related to some analyses on mitochondrial proteins and to the fact that proteomic and transcriptomic comparison may be problematic in catabolic conditions because some gene expression is required to maintain or re-establish enzymes/proteins that are destroyed by the proteolytic systems (including the autophagy proteins and ubiquitin ligases). The authors should consider the following points.Point 1. The authors used the name sarcopenia as synonymous with muscle atrophy. However, sarcopenia clearly defines the disease state (disease code: ICD-10-CM (M62.84)) of excessive muscle loss and force drop during ageing (Ref: Anker SD et al. J Cachexia Sarcopenia Muscle 2016 Dec;7(5):512-514.). Therefore, the word sarcopenia must be used only when pathological age-related muscle loss is the subject of study. Sarcopenia can be present in cancer patients who also experience cachexia, however since the age of tumor-bearing mice in this study is 7-9 weeks old, the authors should refrain from using sarcopenia and instead replace it with the words muscle atrophy/ muscle wasting/muscle loss.

Response: This issue has also been raised by the Reviewer #1. We agree that the term sarcopenia historically refers to aged muscle, but it is also used in oncology (for example, in this article, which we quote: Mintziras I et al. Sarcopenia and sarcopenic obesity are significantly associated with poorer overall survival in patients with pancreatic cancer: Systematic review and meta-analysis. Int J Surg 2018;59:19-26). Actually, the term sarcopenia is now widely used in the literature and in the clinic to describe the loss of muscle mass and strength in cancer patients (see for example, this recent review: Papadopetraki A. et al. The Role of Exercise in Cancer-Related Sarcopenia and Sarcopenic Obesity. Cancers 2023;15;5856).

Point 2. Most of the analyses of mitochondrial function are appropriate. However, the methodological approach to determining mitochondrial fusion and fission machinery shown in Fig. 5F is wrong. The correct way is to normalize the OPA1, MFn1/2 on mitochondrial proteins such as VDAC/porin. In fact, by loading the same amount of total protein (see actin in panel 5F) the difference between a normal and a muscle with enhanced protein breakdown is lost. In fact, we should expect a decrease in actin level in tumor-bearing mice with muscle atrophy while the blots clearly show the same level due to the normalization of protein content. Moreover, by loading the same amount of proteins in the gel, the atrophying muscle lysates become enriched in the proteins/organelles that are less affected by the proteolysis resulting in an artefactual increase. The correct way should be to lyse the whole muscle of control and tumor-bearing mice in an identical volume and to load in western blot the same volume between control cachectic muscles. Alternatively, the relative abundance of mitochondrial shaping proteins related to mitochondrial transmembrane or matrix proteins (mito mass) should compensate for the loading normalization. Because the authors showed elongated mitochondria despite mitophagy genes being up, fragmentation may be altered. Moreover, DNM1l gene is suppressed and therefore DRP1 protein must be analyzed. Finally, OPA 1 protein has different isoforms due to the action of proteases like OMA1, and YME1L that elicit different functions being the long one pro-fusion while the short ones do not. The authors must quantify the long and short isoforms of OPA1.

Response: We acknowledge that our analysis of a minor set of proteins involved in mitochondrial dynamics by Western blotting (Figure 5F) is basic and could have been improved. We thank the Reviewer for all the suggestions, which will be very useful in future projects studying the subject in greater depth and according to the molecular characteristics of each player in mitochondrial fusion, fission, mitophagy and biogenesis.

Point 3. The comparison of proteomic and transcriptomic profiles to identify concordance or not is problematic when atrophy programs are induced. In fact, most of the transcriptional-dependent upregulation is to preserve/maintain/reestablish enzymes that are consumed during enhanced protein breakdown. For instance, the ubiquitin ligases when activated undergo autoubiquitination and proteasome degradation. The same happens for several autophagy-related genes belonging to the conjugation system (LC3, Gabarap), the cargo recognition pathways (e.g. Ubiquitin, p62/SQSTM1) and the selective autophagy system (e.g. BNIP3, PINK/PARKIN) and metabolic enzymes (e.g. GAPDH, lipin). Finally, in case identical amounts of proteins have been loaded in mass spec the issues rise in point 2 of selective enrichment should be considered. Therefore, when comparing proteomic and transcriptomic these issues should be considered in discussion.

Response: We fully agree with the Reviewer that seeking concordance between transcriptomic and proteomic data in the case of an organ affected by a high level of proteolysis is a difficult business. Another major difficulty we discussed in the Discussion section of the article is the fact that there is no concordance between RNA and protein level for a good proportion of proteins, for multiple reasons, so each level of omics has to be interpreted independently to give information on the pathophysiology of the organ studied.